# A single K⁺-binding site in the crystal structure of the gastric proton pump

Kenta Yamamoto[1,2], Vikas Dubey[3], Katsumasa Irie[1,2], Hanayo Nakanishi[1,2], Himanshu Khandelia[3], Yoshinori Fujiyoshi[1,4], Kazuhiro Abe[1,2]*

[1]Cellular and Structural Physiology Institute, Nagoya University, Nagoya, Japan; [2]Graduate School of Pharmaceutical Sciences, Nagoya University, Nagoya, Japan; [3]Department of Physics, Chemistry and Pharmacy, PHYLIFE, University of Southern Denmark, Odense, Denmark; [4]CeSPIA Inc, Tokyo, Japan

**Abstract** The gastric proton pump (H⁺,K⁺-ATPase), a P-type ATPase responsible for gastric acidification, mediates electro-neutral exchange of H⁺ and K⁺ coupled with ATP hydrolysis, but with an as yet undetermined transport stoichiometry. Here we show crystal structures at a resolution of 2.5 Å of the pump in the E2-P transition state, in which the counter-transporting cation is occluded. We found a single K⁺ bound to the cation-binding site of the H⁺,K⁺-ATPase, indicating an exchange of 1H⁺/1K⁺ per hydrolysis of one ATP molecule. This fulfills the energy requirement for the generation of a six pH unit gradient across the membrane. The structural basis of K⁺ recognition is resolved and supported by molecular dynamics simulations, establishing how the H⁺,K⁺-ATPase overcomes the energetic challenge to generate an H⁺ gradient of more than a million-fold—one of the highest cation gradients known in mammalian tissue—across the membrane.

DOI: https://doi.org/10.7554/eLife.47701.001

*For correspondence: kabe@cespi.nagoya-u.ac.jp

## Introduction

The highly acidic environment (pH 1) in the stomach is generated by the gastric proton pump, H⁺, K⁺-ATPase, which mediates an exchange of H⁺ and K⁺ across the parietal cell membrane that is coupled with ATP hydrolysis (*Figure 1*) (*Ganser and Forte, 1973*). Like other P-type ATPases, the vectorial cation transport of gastric H⁺,K⁺-ATPase is accomplished by cyclical conformational changes in the enzyme (abbreviated as 'E') (*Rabon and Reuben, 1990*), generally described using E1/E2 nomenclature based on the Post-Albers scheme for Na⁺,K⁺-ATPase (*Figure 2A*) (*Post et al., 1969*). During the transport cycle, a conserved aspartate is reversibly auto-phosphorylated to form phosphoenzyme intermediates (EPs), a hallmark of members of the P-type ATPase family (*Post and Kume, 1973*). The H⁺,K⁺-ATPase consists of two subunits. The 110 kDa catalytic α-subunit is homologous to other related P2-type ATPases such as Na⁺,K⁺-ATPase, with which it shares 65% identity (*Morth et al., 2007*), and the serco(endo)plasmic reticulum Ca²⁺-ATPase (SERCA) (*Toyoshima et al., 2000*), with which it shares 35% sequence identity (*Palmgren and Axelsen, 1998*). The α-subunit is composed of 10 transmembrane (TM) helices, in which the cation-binding sites are located, and three cytoplasmic domains (the nucleotide (N), phosphorylation (P) and actuator (A) domains) that catalyze ATP hydrolysis and the auto-phosphorylation reaction. The β-subunit is a single-span membrane protein with a short N-terminal cytoplasmic tail and a large C-terminal ectodomain, and it is involved in correct membrane integration and targeting of the complex to the cell surface (*Chow and Forte, 1995*).

Although the closely related Na⁺,K⁺-ATPase mediates electrogenic transport of three Na⁺ and two K⁺ ions coupled with the hydrolysis of one ATP molecule (*Jorgensen et al., 2003*), the number of transported cations in the electroneutral operation (*Sachs et al., 1976*; *van der Hijden et al.,*

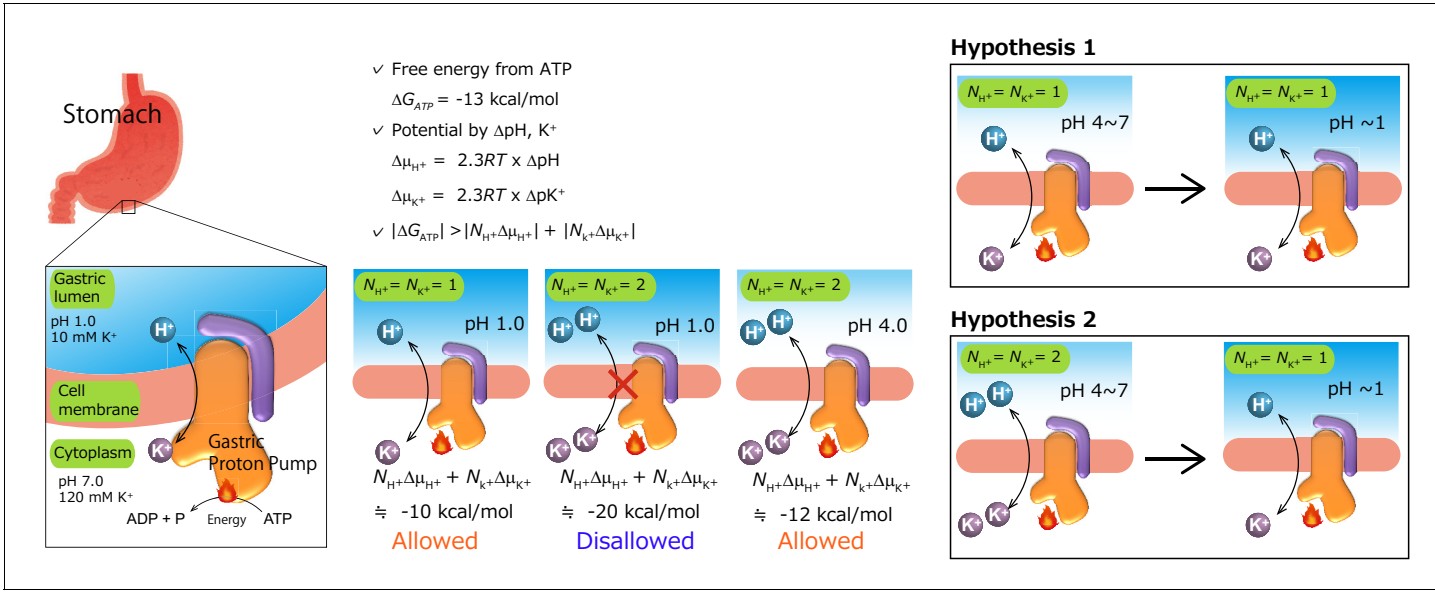

**Figure 1.** Transport stoichiometry and free energy from ATP hydrolysis. The free energy that is derived from ATP hydrolysis, $\Delta G_{ATP}$—calculated from $\Delta G'_0$ and the measured intracellular concentrations of ATP, ADP and $P_i$ in the parietal cell—is about $-13$ kcal/mol (**Durbin et al., 1974**). Under physiological conditions in intact parietal cells with an internal pH of approximately 7, a pH gradient of at least 6 pH units must be created. The maximum electrochemical gradient, $\Delta\mu_i$, that can be formed by an ion-transporting ATPase is a function of the free energy of ATP hydrolysis. Taking reasonable values of pH 7 and 120 mM $K^+$ for intracellular conditions and the measured pH (1.0) and $K^+$ concentration (10 mM) of the gastric juice, we can calculate concentration gradients across the parietal cell membrane of $10^6$ and 12 times for $H^+$ and $K^+$, respectively. For an electro-neutral $H^+$/$K^+$ exchange pump for which $N_{H+}=NK_{K+}=1$, the sum of chemical potentials is about $-10$ kcal/mol, and is within the range of ATP free energy (**Reenstra and Forte, 1981**). However, if the exchange of two cations is assumed, in which case $N_{H+}=NK_{K+}=2$, the reaction is thermodynamically disallowed at pH 1. Therefore, the ratio of $H^+$ transported to ATP hydrolyzed must be approximately 1, and cannot be as large as 2, when gastric pH is around 1. However, this cannot be the case when luminal pH is neutral to weakly acidic (e.g., at pH 4). A different postulate is that two $H^+$ are transported for each ATP molecule hydrolyzed under these conditions (**Rabon et al., 1982**), and that the number of transported $H^+$, and therefore $K^+$ as well, changes from 2 to 1 as luminal pH decreases. On the basis of two different results for $H^+$/ATP ratio, two hypotheses have been proposed. Hypothesis 1: transport stoichiometry remains constant (1$H^+$:1$K^+$) regardless of luminal pH. Hypothesis 2: 2$H^+$:2$K^+$ ions are exchanged when luminal pH is neutral to weakly acidic, and it returns to 1$H^+$:1$K^+$ transport mode when the luminal solution becomes highly acidic.
DOI: https://doi.org/10.7554/eLife.47701.002

**1990**; **Burnay et al., 2001**; **Burnay et al., 2003**) of the gastric $H^+,K^+$-ATPase remains unclear. Under physiological conditions, intact parietal cells with an internal pH of approximately seven must generate a pH gradient of at least six pH units (**Wolosin, 1985**). The secretion of a relatively voluminous flow of gastric acid requires the expenditure of considerable cellular energy. Taking reasonable values of pH 7 and 120 mM $K^+$ for the intracellular condition, together with measured pH (1.0) and $K^+$ concentration (10 mM) of the gastric juice, we can calculate concentration gradients across the membrane of $10^6$ and 12 times for $H^+$ and $K^+$, respectively. The sum of chemical potentials is about $-10$ kcal/mol. The reported free energy derived from ATP hydrolysis in the parietal cell is about $-13$ kcal/mol (**Durbin et al., 1974**). Therefore, the ratio of $H^+$ transported to ATP hydrolyzed must be approximately 1, and cannot be as large as 2, when the gastric pH approaches 1 (**Figure 1**). Accordingly, a previous investigation of $H^+$ transport shows a 1:1 stoichiometry of $H^+$ transport and ATP hydrolysis at pH 6.1–6.9 (**Reenstra and Forte, 1981**). However, this ratio might change when the luminal pH is neutral to weakly acidic, in which case a 2$H^+$:1ATP stoichiometry would be feasible in principle (**Figure 1**). A separate investigation, on the other hand, showed the transport of two $H^+$'s for each ATP hydrolyzed at pH 6.1 (**Rabon et al., 1982**). It was speculated that the number of $H^+$ transported may change from two (at neutral pH) to one as the luminal pH decreases (**Shin et al., 2009**; **Abe et al., 2012**). The electroneutrality of the transport cycle of the $H^+,K^+$-ATPase over a wide pH range is indisputable based on the electrophysiological studies (**Sachs et al., 1976**; **van der Hijden et al., 1990**; **Burnay et al., 2001**; **Burnay et al., 2003**). Therefore, whether the $H^+,K^+$-ATPase constantly exchanges 1$H^+$:1$K^+$ regardless of luminal pH (**Figure 1**, Hypothesis 1) or switches

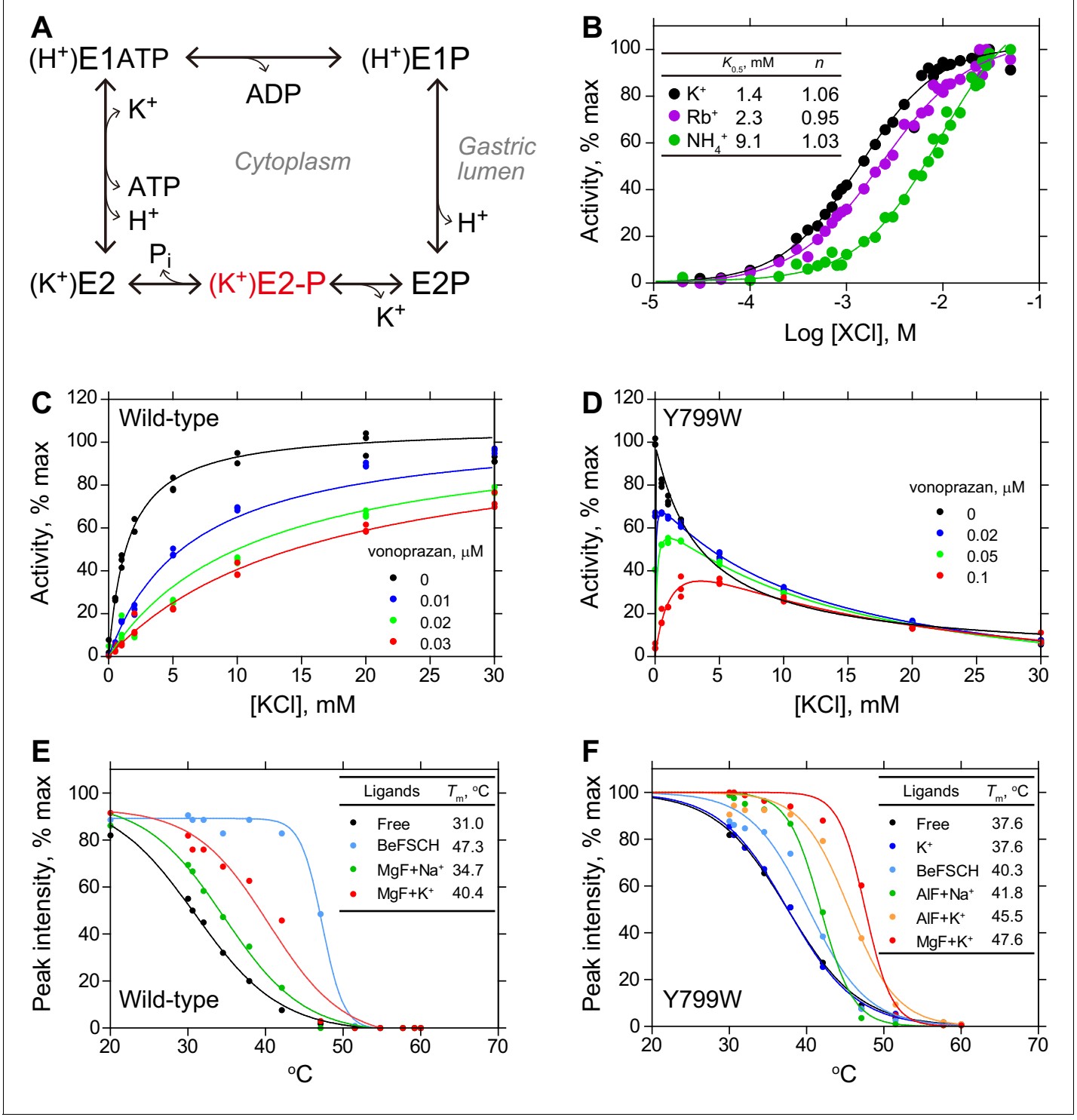

**Figure 2.** Characterization of the Tyr799Trp mutant of H⁺,K⁺-ATPase. (A) Post-Albers type reaction scheme for H⁺,K⁺-ATPase. The K⁺-occluded E2-P transition state is highlighted in red. (B) ATPase activities of the wild-type enzyme with the indicated cations, showing a Hill coefficient (n) close to 1. K⁺-dependent ATPase activity of (C) wild-type or (D) Tyr799Trp mutant H⁺,K⁺-ATPase in the absence or presence of indicated concentrations of the K⁺-competitive inhibitor vonoprazan. Thermal stability of (E) wild-type or (F) Try799Trp mutant H⁺,K⁺-ATPase in the presence of the indicated ligands, evaluated by fluorescence size-exclusion chromatography.

DOI: https://doi.org/10.7554/eLife.47701.003

its transport mode from $1H^+:1K^+$ to $2H^+:2K^+$ depending on conditions (*Figure 1*, Hypothesis 2) has long time been the subject of debate in the membrane transport field. The reason for the discrepancy probably lies in the difficulty in measuring the transport stoichiometry, namely, one $H^+$ or two $H^+$'s per ATP hydrolysis. These measurements are very sensitive because the loss of $H^+$ from the external space of the vesicle must be measured with a glass electrode in vitro, and scalar proton production resulting from ATP hydrolysis must be corrected for. The corrected rate of $H^+$ transport is then compared with that of ATP hydrolyzed under the same conditions as an independent measurement. The inside-out vesicles used in these experiments are sedimented from the natural source (i. e., pig stomach), and contain $H^+,K^+$-ATPase as approximately 70% of their total protein (*Abe and Olesen, 2016*). This system may introduce a different estimate of ATPase activity, especially with regards to the interpretation of the basal $Mg^{2+}$-sensitive ATPase fraction in the absence of $K^+$. Therefore, we reason that the best way to resolve the disputed transport stoichiometry of the pump is by direct observation of the number of counter-transported cations occluded in the $H^+,K^+$-ATPase at neutral pH, given that the transport is approximately electroneutral (*Sachs et al., 1976*; *van der Hijden et al., 1990*; *Burnay et al., 2003*; *Burnay et al., 2001*). To this end, we determined crystal structures at neutral pH, in order to determine the maximum capacity of $K^+$ occlusion.

## Results

### Tyr799Trp mutant $H^+,K^+$-ATPase prefers the $K^+$-occluded E2-P transition state

According to the transport cycle of the $H^+,K^+$-ATPase (*Rabon and Reuben, 1990*) (*Figure 2A*), binding of counter-transporting $K^+$ induces luminal gate closure and accelerates dephosphorylation of the auto-phosphorylated intermediate, E2P. Subsequently, the enzyme moves to the E2-P transition state in which counter-transported $K^+$ is occluded (*Abe et al., 2012*). For crystallization, the transition state phosphate analogs (*Danko et al., 2004*) (magnesium fluoride ($MgF_x$) or aluminum fluoride ($AlF_x$)) together with counter-transporting cation ($K^+$ or its congener $Rb^+$, see *Figure 2B*) were applied to the wild-type (WT) $H^+,K^+$-ATPase. However, the crystals diffracted poorly in these conditions and the resolution was limited to 4.3 Å, which is insufficient for the precise definition of the $K^+$-coordination. Better-resolving crystals were obtained by using a Tyr799Trp mutation on the α-subunit (Y799W). This mutant was originally obtained from the screening of the inhibitor-binding site of $H^+,K^+$-ATPase, and its unusual ATPase profile prompted us to apply this mutant to the crystallization of the $K^+$-occluded form.

Tyr799 is located at the entrance of the luminal-facing cation gate to which the specific inhibitors bind (*Abe et al., 2018*), 12 Å distant from the $K^+$ binding site, and its mutation does not affect the $K^+$ binding site directly. To our surprise, however, the Tyr799Trp mutant shows its highest ATPase activity in the absence of $K^+$ and the activity decreases with increasing $K^+$ concentration, in marked contrast to the $K^+$-dependent increase in ATPase activity of the wild-type enzyme (*Figure 2C,D*). The observed $K^+$-independent ATPase activity of Tyr799Trp in the absence of $K^+$ indicates that the luminal gate of this mutant is spontaneously closed, like those of the previously reported constitutively active mutants (*Abe et al., 2018*). The Tyr799Trp mutation exerts a molecular signal that induces E2P dephosphorylation when $K^+$ is bound to the cation binding site. Decreasing ATPase activity of Tyr799Trp with increasing $K^+$ concentration ($K_{0.5,cyto}$ = ~5 mM) can be interpreted as the result of $K^+$-occlusion from the cytoplasmic side, which is also observed in the wild-type enzyme, albeit with much lower affinity ($K_{0.5,cyto}$ =~200 mM) (*Ljungström et al., 1984*). Furthermore, in the presence of an inhibiting concentration of the $K^+$-competitive blocker vonoprazan, Tyr799Trp shows high-affinity $K^+$-activation of ATPase activity. These data indicate that, despite the unique properties of luminal gate closure, the cation-binding site of Tyr799Trp is intact and capable of high-affinity $K^+$-binding. The thermal stability (*Hattori et al., 2012*) of Tyr799Trp is significantly higher in the presence of $K^+$ and $AlF_x$ than in the presence of $AlF_x$ alone (*Figure 2F*), qualitatively indicating $K^+$-occlusion in the Tyr799Trp mutant. This $K^+$-occluded E2-P transition state is the most stable conformation of all of the evaluated conditions for Tyr799Trp, whereas for the wild-type enzyme, inhibitor-bound and luminal-open E2BeF is the most stable condition (*Figure 2E*). We therefore conclude that Tyr799Trp prefers the luminal-closed $K^+$-occluded state, $(K^+)$E2-P, and is thus suitable for the structural analysis of the $K^+$-occluded form. As a note, a similar mutant (Phe788Leu in $Na^+,K^+$-

ATPase, which corresponds to Tyr799 in $H^+,K^+$-ATPase) that shows $K^+$-independent dephosphorylation has been reported for the $Na^+,K^+$-ATPase (*Vilsen, 1999*), suggesting the conservation of a luminal gating mechanism between the two related ATPases.

## Crystal structures reveal the number of $K^+$ occluded at the cation-binding site

In order to define the number of $K^+$ that are occluded, and their coordination chemistry in the cation-binding site of the $H^+,K^+$-ATPase, we attempted to obtain a high-resolution structure using the Tyr799Trp mutant. As expected, the crystals were significantly improved when using the Tyr799Trp

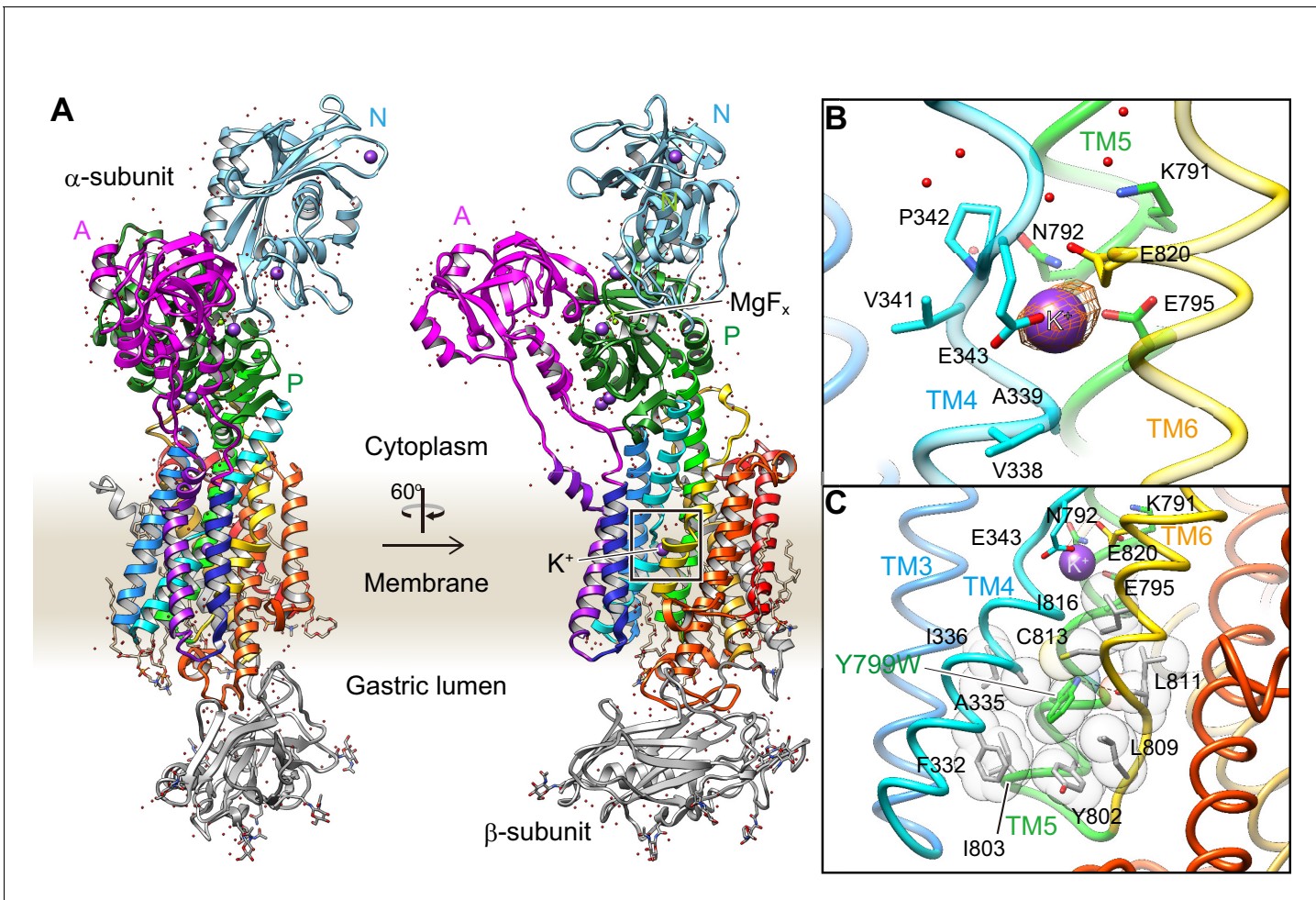

**Figure 3.** Crystal structure of the $K^+$-occluded E2-P transition state of $H^+,K^+$-ATPase. (**A**) Overall structure of the $K^+$-occluded E2-MgF$_x$ state [Y799W($K^+$) E2-MgF$_x$] in ribbon representations. For the α-subunit, the three cytoplasmic domains (A, P and N) are shown in different colors. The color of the TM helices gradually changes from purple to red (TM1–TM10). The β-subunit with a single TM helix and six *N*-glycosylation sites in the ecto-domain is shown in gray. Phospholipids, a cholesterol and detergent molecules are also modeled (as sticks). Red dots and purple spheres represent water molecules and $K^+$ ions, respectively. (**B**) The TM $K^+$-binding site viewed from a position parallel to membrane plane. Orange mesh represents an anomalous density map of the $Rb^+$-bound form [($Rb^+$)E2-MgF$_x$] with 8 σ contour level. Amino acids that contribute to the $K^+$-coordination are shown in sticks. (**C**) The hydrophobic gate centered around Tyr799Trp (green), with surrounding hydrophobic residues (gray), is shown. The dotted line indicates a hydrogen bond between a nitrogen atom of the Trp residue and a main chain oxygen of L811.

DOI: https://doi.org/10.7554/eLife.47701.004

The following figure supplements are available for figure 3:

**Figure supplement 1.** Comparison of the molecular conformations of the $K^{+-}$ or $Rb^+$-occluded E2-P transition state of different $H^+,K^+$-ATPases.
DOI: https://doi.org/10.7554/eLife.47701.005

**Figure supplement 2.** Crystal structure of the wild-type enzyme.
DOI: https://doi.org/10.7554/eLife.47701.006

mutant, which provided a 2.5 Å resolution structure in the best case [Y799W(K$^+$)E2-MgF$_x$] (**Figure 3**). We analyzed several crystal structures in the presence of different combinations of K$^+$ or Rb$^+$, and AlF$_x$ or MgF$_x$, all of which mimic the K$^+$-occluded E2-P transition state and are indistinguishable in molecular conformation (**Figure 3—figure supplement 1**). Although the analyzed resolution is limited, the structure of the wild-type enzyme also shows almost the same molecular conformation as that of the Tyr799Trp mutant (**Figure 3—figure supplement 2**). We therefore use the Y799W(K$^+$)E2-MgF$_x$ structure analyzed at the best resolution in the following discussion (**Table 1**). The overall structure of H$^+$,K$^+$-ATPase Y799W(K$^+$)E2-MgF$_x$ (**Figure 3A**) is very close to the corresponding structure of Na$^+$,K$^+$-ATPase (**Morth et al., 2007**; **Shinoda et al., 2009**) (**Figure 3—figure supplement 1**). However, instead of having two K$^+$ ions occluded in the transmembrane cation-binding site of the Na$^+$,K$^+$-ATPase, only a single K$^+$ is observed in the cation-binding site of H$^+$,K$^+$-ATPase (**Figure 3**). The observed single K$^+$-binding is confirmed by the anomalous difference Fourier maps of Y799W(Rb$^+$)E2-MgF$_x$ (**Figure 3B**), Y799W(Rb$^+$)E2-AlF$_x$ and Y799W(K$^+$)E2-MgF$_x$ structures (**Figure 3—figure supplement 1**), which unambiguously show a single strong peak at the cation binding site located between TM4, TM5 and TM6 in the middle section of the membrane. The presence of saturating concentrations of cation (400 mM KCl or RbCl) in the crystallization buffer ensures high occupancy of K$^+$ at the cation-binding site, although several other binding sites, presumably low-affinity and/or non-specific, were determined in the cytoplasmic domains (**Figure 3—figure supplement 1**).

**Table 1.** Data collection and refinement statistics.

| | Y799W(K$^+$)E2-MgF$_x$ | Y799W(Rb$^+$)E2-MgF$_x$ | Y799W(Rb$^+$)E2-AlF$_x$ | WT(Rb$^+$)E2-MgF$_x$ |
|---|---|---|---|---|
| **Data collection** | | | | |
| Resolution (Å)[†] | 2.7 × 2.8 × 2.5 (2.6–2.5)[‡] | 2.8 × 2.8 × 2.6 (2.7–2.6) | 2.8 × 3.0 × 2.5 (2.6–2.5) | 5.1 × 5.1 × 4.3 (4.5–4.3) |
| Space group | P 3$_1$ 2 1 | P 3$_1$ 2 1 | P 3$_1$ 2 1 | C 1 2 1 |
| **Cell dimensions** | | | | |
| a, b, c (Å) | 103.37, 103.37, 370.01 | 103.38, 103.38, 369.86 | 103.23, 103.23, 369.44 | 191.51, 106.43, 250.96 |
| α, β, γ (°) | 90, 90, 120 | 90, 90, 120 | 90, 90, 120 | 90, 107.79, 90 |
| $R_{merge}$ | 0.067 (3.685) | 0.080 (2.248) | 0.106 (3.499) | 0.152 (1.359) |
| $R_{pim}$ | 0.022 (1.179) | 0.030 (0.855) | 0.040 (1.295) | 0.065 (0.544) |
| I/σI | 20.18 (0.93) | 16.19 (1.16) | 11.64 (0.77) | 7.19 (0.92) |
| C/C1/2 | 0.99 (0.59) | 0.99 (0.70) | 0.99 (0.60) | 0.99 (0.75) |
| Completeness (%) | 85.65 (26.79) | 88.93 (36.11) | 76.00 (18.98) | 94.94 (53.46) |
| Redundancy | 10.2 (10.6) | 8.0 (7.8) | 8.0 (8.2) | 6.6 (7.2) |
| **Refinement** | | | | |
| Resolution (Å) | 48–2.5 (2.6–2.5) | 48–2.6 (2.7–2.6) | 50–2.5 (2.6–2.5) | 48–4.3 (4.5–4.3) |
| No. of reflections | 69,037 (2135) | 63,831 (2533) | 61,013 (1502) | 31,390 (1747) |
| $R_{work}$/$R_{free}$ (%) | 20.1/25.7 (37.3/48.0) | 20.6/26.2 (35.1/43.5) | 21.5/27.4 (32.7/42.6) | 26.3/33.9 (33.4/41.2) |
| Wilson B-factor | 60.74 | 52.60 | 48.43 | 158.52 |
| No. of atoms | 10,513 | 10,485 | 10,498 | 19,823 |
| Protein | 9736 | 9762 | 9763 | 19,623 |
| Ligands | 362 | 362 | 364 | 200 |
| Average B-factor | 74.40 | 62.68 | 60.03 | 244.07 |
| Protein (Å$^2$) | 72.62 | 60.98 | 57.83 | 243.40 |
| Ligands (Å$^2$) | 122.80 | 114.14 | 103.16 | 310.29 |
| **R.m.s deviations** | | | | |
| Bond lengths (Å) | 0.010 | 0.009 | 0.010 | 0.005 |
| Bond angles (°) | 1.36 | 1.51 | 1.39 | 1.18 |

[†]The diffraction data are anisotropic. The resolution limits given are for the $a^*$, $b^*$ and $c^*$ axes, respectively.

[‡]Statistics for the highest-resolution shell are shown in parentheses.

DOI: https://doi.org/10.7554/eLife.47701.007

To exclude the possibility that the observed single $K^+$-binding results from an artifact of the Tyr799Trp mutation, we determined the wild-type WT(Rb$^+$)E2-MgF$_x$ structure in the same conformation (*Figure 3—figure supplement 2*), despite its limited resolution of 4.3 Å. The β-subunit ectodomain in protomer A, and the A- and N-domains in protomer B show relatively weak density, due to lack of tight crystal packing at these regions. The N-terminal tail of the β-subunit is partially visible in the wild-type structure, in contrast to that in the Tyr799Trp structure. We could only assign an additional nine amino acids (Tyr20–Gln28 of the β-subunit) for this region that extends along with the membrane surface. The morphology of the β-subunit N-terminus is similar to that observed in the previously reported electron crystallographic structures (*Abe et al., 2009*). However, the well-ordered regions, especially for the TM helices that define the luminal-closed $K^+$-occluded state, are almost identical to those of the high-resolution Tyr799Trp structure. We therefore conclude that the molecular conformation of the wild-type H$^+$,K$^+$-ATPase is essentially the same as that of the Tyr799Trp mutant. At the cation-binding site of the wild-type, a single Rb$^+$ anomalous peak is found at the position close to the bound $K^+$ in Tyr799Trp, as well as at site II in Na$^+$,K$^+$-ATPase (*Morth et al., 2007*; *Shinoda et al., 2009*). An anomalous signal is hardly seen at the position corresponding to the cation-binding site I in Na$^+$,K$^+$-ATPase, even at the low contour level (*Figure 3—figure supplement 2*). These observations allow us to assert that the conclusions extracted for the Tyr799Trp mutant in regard to $K^+$ stoichiometry can be extended to the wild-type protein.

A single $K^+$-binding in H$^+$,K$^+$-ATPase is also supported by a Hill coefficient for $K^+$ of close to 1.0 (*Figure 2B*), in marked contrast to the Hill coefficient of 1.5 for the $K^+$-dependence of Na$^+$,K$^+$-ATPase (*Sweadner, 1985*) in which two $K^+$ ions are occluded at the cation-binding site. Crystals were generated at the close to neutral pH of 6.5, the condition also used for previous in vitro H$^+$ transport measurements (*Reenstra and Forte, 1981*; *Rabon et al., 1982*). Assuming electro-neutral transport in the H$^+$,K$^+$-ATPase (*Sachs et al., 1976*; *van der Hijden et al., 1990*; *Burnay et al., 2003*; *Burnay et al., 2001*), if two H$^+$'s are transported at neutral pH, as suggested by a previous in vitro measurements (*Rabon et al., 1982*), then two $K^+$ ions must be counter-transported. A single bound $K^+$ observed in the crystal structure therefore strongly indicates that H$^+$,K$^+$-ATPase transports only one $K^+$ ion at once, and therefore only one H$^+$ in the opposite direction, for every ATP hydrolyzed at neutral pH (*Figure 1*, Hypothesis 1) (*Reenstra and Forte, 1981*).

## Mechanism for the luminal gate closure

Unlike the previously reported Rb$^+$-bound and luminal-open (SCH)E2BeF$_x$ structure of H$^+$,K$^+$-ATPase (*Abe et al., 2018*), in the Y799W(K$^+$)E2-MgF$_x$ state, the cation-binding site is tightly shielded from the luminal solution by an extensive hydrophobic cluster centered around Tyr799Trp (*Figure 3C*) at the luminal gate. Substitution of Tyr799 to Trp enhances hydrophobic interactions with the surrounding amino acids, the interaction of which may be strong enough for spontaneous luminal gate closure, as suggested by the high ATPase activity of the Tyr799Trp mutant in the absence of $K^+$ (*Figure 2*, *Figure 4*). A hydrogen bond between the nitrogen of the Trp799 residue and main chain carbonyl oxygen of Leu811 (3.3 Å, *Figure 3C*, *Figure 4A*) provides for the positioning of a Trp rotamer suitable for hydrophobic interactions with its surroundings, and stabilizes its conformation. Molecular dynamics (MD) simulations support the presence of this hydrogen bond, and also show that in the wild-type enzyme, the hydroxyl group of Tyr799 makes a similar hydrogen bond with the main chain carbonyl oxygen of Leu811 (3.0 Å, *Figure 4B*, *Video 1*). However, the side chain of Tyr is smaller than that of Trp, and thus forms a weaker hydrophobic network with the surrounding residues. These differences are reflected in the stability of the luminal gate conformation in the MD simulations, as seen in the larger fluctuation of Tyr799 in the wild-type when compared to that in the Tyr799Trp mutant (*Figure 4C*, *Video 1*).

To confirm the effect of a hydrogen bond between Tyr799Trp and main chain carboxyl oxygen of Leu811, and hydrophobic interactions with surrounding residues on luminal gate stability, we conducted mutational analysis. Although the Leu811Pro mutant is inactive, probably because it is unfolded during expression (as evaluated by fluorescence size-exclusion chromatography), the Leu811Gly mutant in the background of Tyr799Trp shows weak $K^+$-dependence in its ATPase activity (*Figure 4*). This evidence suggests that the glycine substitution might alter the main-chain trace near Leu811, thus weakening the hydrogen-bond interaction with Trp799. Hydrophilic serine substitution of the surrounding hydrophobic residues in the Try799Trp background does not significantly change the inverse $K^+$-dependence profile of Tyr799Trp activity, except that the Tyr799Trp + Ile803Ser

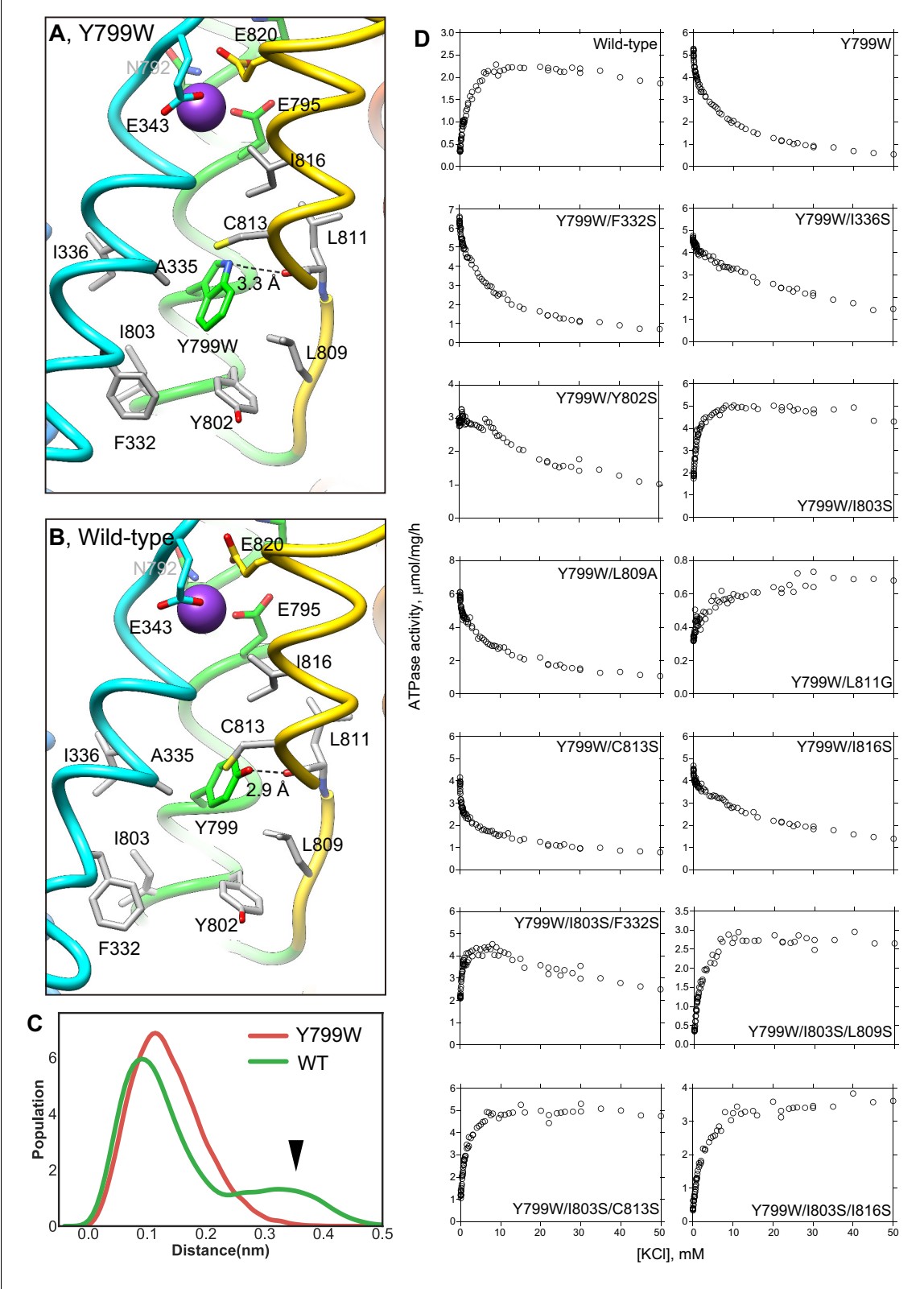

**Figure 4.** K$^+$-dependence on ATPase activities of Y799W and mutation of the surrounding hydrophobic amino acids. Hydrophobic interactions observed in the luminal gate of Y799W (A) and WT (B) H$^+$,K$^+$-ATPases. Hydrophobic residues that are likely to contribute to the luminal gate stability are shown as gray sticks. (C) Distribution of the root-mean square deviations (RMSD) for NH group of the Trp residue in Y799W (red line, mean RMSD is 1.23 ± 0.04) and the OH group of the Tyr799 residue in the WT H$^+$,K$^+$-ATPase (green line, 1.59 ± 0.27). The arrowhead points to a population of large

*Figure 4 continued on next page*

*Figure 4 continued*

RMSD, indicating higher fluctuations of the OH group in the WT enzyme compared to those of the NH group in the Tyr799Trp mutant (see also *Video 1*). (D) $K^+$-dependence of the vonoprazan-sensitive ATPase activities of the indicated mutants. Data plotted had background values in the presence of 10 µM vonoprazan subtracted. Individual data from a 96-well plate at different concentrations of $K^+$ were plotted, and representative results from more than three independent measurements for each mutant are shown in the figure.

DOI: https://doi.org/10.7554/eLife.47701.009

The following figure supplement is available for figure 4:

**Figure supplement 1.** Conformational change upon $K^+$-occlusion.

DOI: https://doi.org/10.7554/eLife.47701.010

mutant (Y799W/I803S) shows a $K^+$-dependent increase in its ATPase activity. However, a $K^+$-independent ATPase fraction remains in the absence of $K^+$. In the background of Y799W/I803S, an additional third mutation (Leu809Ser, Cys813Ser or Ile816Ser) restores $K^+$-dependent ATPase activity to a level approximating that of the wild-type enzyme. These data suggest that spontaneous gate closure of the Tyr799Trp mutant is caused by the hydrophobic interactions with its surroundings, and that these interactions are facilitated by the favorable rotamer position of Tyr799Trp guided by the hydrogen-bond between Trp799 and Leu811 main chain. Wild-type-like $K^+$-dependence can be restored by additional mutagenesis based on the observed structure of Y799W($K^+$)E2-P. We therefore conclude that the luminal-closed molecular conformation that is spontaneously induced by the Try799Trp mutation is not an artifact, and that the driving force for the gate closure is essentially the same as that in the wild-type enzyme.

Comparison of the luminal-open E2P ground state [(von)E2BeF$_x$ structure with bound vonoprazan, a specific inhibitor for H$^+$,K$^+$-ATPase (*Abe et al., 2018*)] and the $K^+$-occluded and luminal-closed E2-P transition state [Y799W($K^+$)E2-MgF$_x$ structure] reveals several key conformational rearrangements upon luminal gate closure (*Figure 4—figure supplement 1*). Gate closure brings the luminal portion of TM4 (TM4L) close to TM6, effectively capping the cation-binding site from the luminal side of the membrane. This lateral shift of TM4L is coupled to the vertical movement of the TM1,2 helix bundle connected to the A domain, leading to the ~30° rotation of the A domain that induces dephosphorylation of the aspartylphosphate. The molecular events required for the luminal gate closure have been extensively studied in SERCA (*Olesen et al., 2004*; *Toyoshima et al., 2007*; *Toyoshima, 2009*), and the same mechanism is observed in the H$^+$,K$^+$-ATPase, confirming the low-resolution maps of electron crystallography (*Abe et al., 2011*; *Abe et al., 2014*). The lateral shift of TM4L not only blocks the physical path of the cation from the luminal solution, but also brings main chain oxygen atoms that are important for the high-affinity $K^+$-coordination to their optimal positions, as described later.

## K$^+$-binding site

How the protein recognizes its specific transport substrate is one of the central questions for membrane transport proteins. Our crystal structure defines a high-affinity $K^+$-binding site of H$^+$, K$^+$-ATPase (*Figure 5*, *Video 2*), with the coordination geometry of $K^+$ and the surrounding amino acids evident at 2.5 Å resolution. The bound single $K^+$ in H$^+$,K$^+$-ATPase is located at a position corresponding to site II of the Na$^+$,K$^+$-ATPase (2K$^+$)E2-MgF$_x$ state (*Morth et al., 2007*; *Shinoda et al., 2009*). The bound $K^+$ is coordinated by eight oxygen atoms located within 4 Å (*Table 2*). Of these, five make a large contribution to $K^+$ coordination (within 3 Å); they include three oxygen atoms from main-chain carbonyls (Val338, Ala339 and Val341) and two from side-chain carboxyl groups (Glu343 and Glu795). The total valence (*Kanai et al., 2013*; *Brown and*

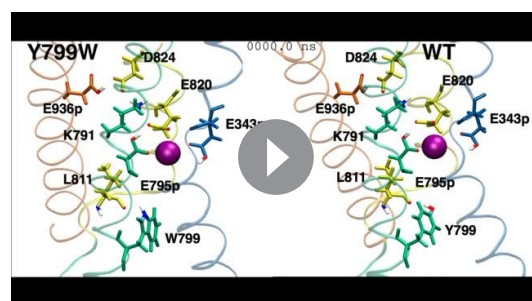

**Video 1.** MD simulation of the Tyr799Trp mutant and wild-type enzymes. A movie shows the luminal gate and $K^+$-binding site structure of 100 ns MD simulations for Tyr799Trp mutant (left) and the wild-type enzyme.

DOI: https://doi.org/10.7554/eLife.47701.008

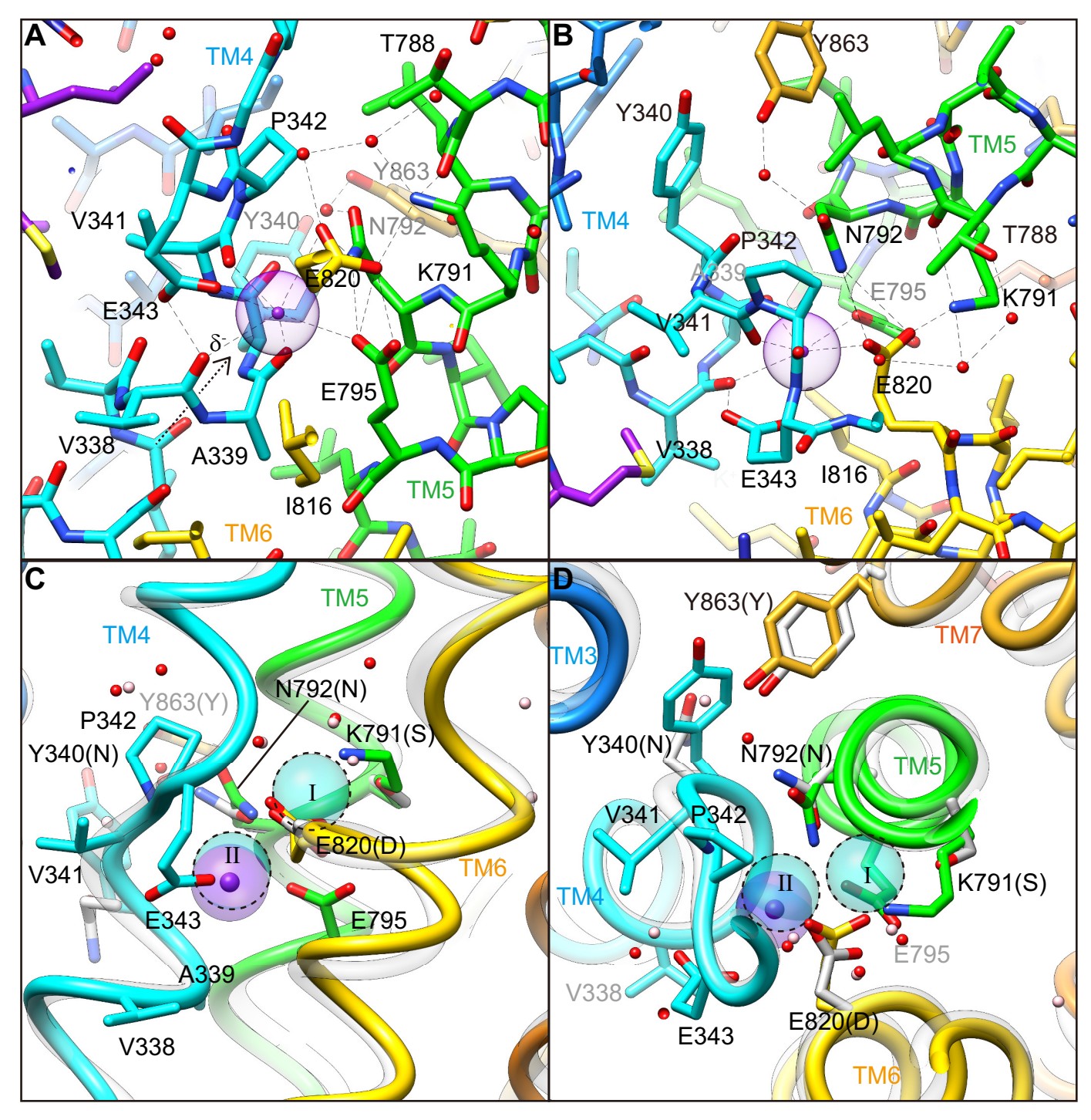

**Figure 5.** K$^+$-binding site. (**A, B**) Close-ups of the transmembrane cation-binding site in the H$^+$,K$^+$-ATPase Y799W(K$^+$)E2-MgF$_x$ state in stick representation, viewed approximately parallel to the membrane from the TM6 side (**A**) or approximately perpendicular to the membrane from the cytoplasmic side (**B**). Dotted lines indicate atoms that are within 3.5 Å of neighboring atoms, presumably forming hydrogen bonds or electrostatic interactions. The purple dot indicates bound K$^+$ with its Stokes radius shown as a transparent sphere. Water molecules (red dots) are also indicated. (**C, D**) The (2K$^+$)E2-MgF$_x$ state of Na$^+$,K$^+$-ATPase (*Shinoda et al., 2009*) (gray ribbons) is superimposed on the corresponding reaction state of H$^+$,K$^+$-ATPase Y799W(K$^+$)E2-MgF$_x$ (colored ribbons), viewed from the membrane (**C**) or cytoplasmic (**D**) sides. For clarity, amino acid residues that contribute to the K$^+$ coordination in H$^+$,K$^+$-ATPase are indicated, with their corresponding amino acids in Na$^+$,K$^+$-ATPase in parentheses. Residues Asn331, Ser782, Asn783, Asp811 and Tyr854 in Na$^+$,K$^+$-ATPase (with corresponding amino acid numbers of Tyr340, Lys791, Asn792, Glu820 and Tyr863, respectively, in H$^+$,K$^+$-ATPase) are shown because three of these amino acids are not conserved. The rest of these amino acids are conserved but Asn

*Figure 5 continued on next page*

Figure 5 continued

792 in H$^+$,K$^+$-ATPase shows different conformation in the structure. Blue spheres with dotted circles indicate positions of K$^+$ binding sites I and II, and pink dots are water molecules in the Na$^+$,K$^+$-ATPase structure.

DOI: https://doi.org/10.7554/eLife.47701.012

*Wu, 1976*) for the bound K$^+$ is 1.05 (optimal valence for K$^+$ is 1.00), indicating that the K$^+$ is almost ideally coordinated by the surrounding oxygen atoms (*Table 2*). Similar results were obtained for Rb$^+$ coordination in the Y799W(Rb$^+$)E2-MgF$_x$ (valence 1.14) and Y799W(Rb$^+$)E2-AlF$_x$ (valence 1.07) structures, but these results are clearly different from those previously reported for the Rb$^+$-bound (SCH)E2BeF$_x$ structure (*Abe et al., 2018*) (valence 0.39, see *Table 2*). As the TM4 helix is unwound at Pro342, coordination by the main chain carbonyl groups becomes possible (*Figure 5*). These carbonyls, in addition to the Glu343 carboxyl, probably determine the positioning of TM4L upon luminal gate closure coupled with K$^+$-occlusion (*Figure 4—figure supplement 1*). It seems to be a natural consequence that these carbonyls dominate K$^+$ coordination, as is also found in Na$^+$,K$^+$-ATPase (*Shinoda et al., 2009*) and K$^+$-channels (*Doyle et al., 1998*). Unwinding of TM4 also provides a negative dipole moment ($\delta^-$) at the K$^+$ site, which is located on the extension of TM4L at the helix breaking point.

Side chain oxygens from Glu343 and Glu795 also contribute to K$^+$ coordination. However, even in the presence of a positive charge (K$^+$ or Rb$^+$), the Glu343 side chain is not attracted to the bound cation. The charge-conserved mutant of Glu343Asp shows no detectable ATPase activity because of its inability to induce K$^+$-dependent dephosphorylation (*Koenderink et al., 2004*), indicating that having a negative charge at this position is not essential for the K$^+$-coordination. Glu343Gln reduces K$^+$-affinity (*Abe et al., 2018*), but to a far lesser extent than Glu343Asp, suggesting that Glu343 is likely to be protonated in the crystal structure. The short distance between the Glu343 carboxyl and Val341 carbonyl (2.6 Å) suggests protonation of Glu343. The juxtaposition of Glu795 and Glu820 indicates that at least one of the carboxyls must be protonated (*Abe et al., 2018*; *Clement et al., 1970*). The almost identical K$^+$-affinity of the wild-type enzyme and the charge-neutralized mutant Glu795Gln suggests that the Glu795 carboxyl is protonated (*Abe et al., 2018*). Therefore, the close location (2.8 Å) of the Glu795 carbonyl, and thus its relatively large contribution to K$^+$-coordination, is not due to a negative charge but to the appropriate size of the Glu side chain, as can be deduced from the reductions in both the ATPase activity and the K$^+$-affinity of the Glu795Asp mutant (*Abe et al., 2018*). The relatively distant location of Glu820 from the bound K$^+$ is due to a salt bridge with Lys791. This counteracts the negative charge at the Glu820 side chain, which is essential for the H$^+$ extrusion into the luminal solution in the previous step of the transport cycle, namely, the luminal-open E2P state (*Abe et al., 2018*).

Thus, K$^+$ is not trapped by the negative charge in the cation-binding site in spite of the crowded space and being surrounded by three glutamate residues. Like water molecules, oxygen atoms derived from either main chain carbonyl or protonated (Glu343 and Glu795) or charge-neutralized (Glu820) acidic side chains in the cation-binding site coordinate K$^+$ through their lone-pair electrons. This idea is consistent with the previously suggested K$^+$-coordination mode in Na$^+$,K$^+$-ATPase (*Yu et al., 2011*; *Rui et al., 2016*; *Cornelius et al., 2018*). Such binding, which is not driven by electrostatic interactions, will aid the release of K$^+$ to the cytoplasmic solution with more than 100 mM K$^+$. Stronger electrostatic interactions between K$^+$ and the side chains would probably prevent the release of K$^+$ even after a conformational change

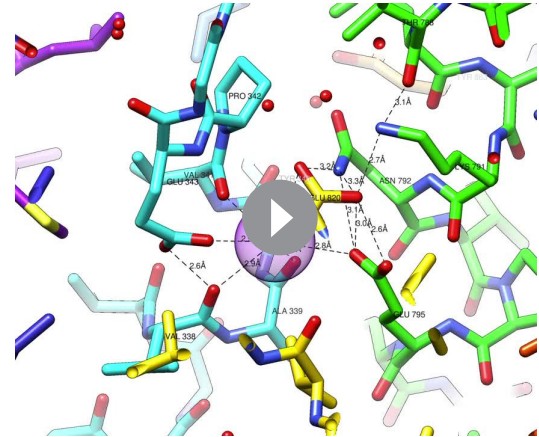

**Video 2.** K$^+$-binding site. Detailed structure of the K$^+$-binding site in the Y799W(K$^+$)E2-MgF$_x$ state, viewed as in *Figure 5A*.

DOI: https://doi.org/10.7554/eLife.47701.011

**Table 2.** Coordination geometry and partial valence in the $K^+$-binding site of $H^+,K^+$-ATPase in $(K^+)$E2-P analogue states.

Only oxygen atoms within 4Å of $K^+$ are included for the valence calculation. Partial valence is calculated for $K^+$ and $Rb^+$ in each corresponding crystal structure.

| Structure | $Rb^+ \cdot (SCH)E2BeF_x$ | | $Y799W(K^+)E2-MgF_x$ | | $Y799W(Rb^+)E2-MgF_x$ | | $Y799W(Rb^+)E2-AlF_x$ | |
|---|---|---|---|---|---|---|---|---|
| Amino acids | Distance (Å) | Valence | Distance (Å) | Valence | Distance (Å) | Valence | Distance (Å) | Valence |
| V338 O | 4.57 | – | 2.90 | 0.11 | 3.00 | 0.12 | 3.09 | 0.1 |
| A339 O | 3.23 | 0.07 | 2.62 | 0.28 | 2.68 | 0.27 | 2.63 | 0.31 |
| V341 O | 2.89 | 0.16 | 2.56 | 0.34 | 2.56 | 0.37 | 2.63 | 0.31 |
| E343 Oε1 | 3.41 | 0.05 | 2.93 | 0.10 | 2.96 | 0.13 | 3.07 | 0.10 |
| E343 Oε2 | 3.42 | 0.05 | 3.74 | 0.01 | 3.63 | 0.03 | 3.72 | 0.03 |
| E795 Oε1 | 4.05 | – | 2.77 | 0.17 | 2.99 | 0.12 | 2.88 | 0.16 |
| E820 Oε1 | 3.56 | 0.04 | 3.32 | 0.03 | 3.21 | 0.08 | 3.38 | 0.05 |
| E820 Oε2 | 3.68 | 0.03 | 3.90 | 0.01 | 3.98 | 0.02 | 3.95 | 0.02 |
| Total valence | | 0.39 | | 1.05 | | 1.14 | | 1.07 |

DOI: https://doi.org/10.7554/eLife.47701.013

is exerted at the cation-binding site in the E1 state.

## Molecular dynamics simulations support protonation states of acidic side chains

Besides the $K^+$-coordinating three glutamates (Glu343, Glu795 and Glu820), there are three other acidic amino acids near the cation-binding site (*Figure 6A,B*). Two of these acidic residues, Asp824 (TM6) and Glu936 (TM8), are located on the opposite side of the Glu795-Glu820 pair with Lys791 in between and are juxtaposed (2.8 Å). One of these acidic side chains is therefore expected to be protonated. Indeed, qualitative estimation of p$K_a$ value using PROPKA (*Li et al., 2005*; *Olsson et al., 2011*; *Søndergaard et al., 2011*) suggests deprotonation of Asp824 (p$K_a$ = 5.4) and protonation of Glu936 (p$K_a$ = 10.8) in the crystal structure. Although the Asp824–Glu936 pair is rather isolated from the $K^+$-binding site in the E2-P form of $H^+,K^+$-ATPase, this acidic residue pair seems to be capable of neutralizing the positive charge of Lys791 when it flips, as is expected in the E1 conformation. The constitutively active ATPase activity in the charge-neutralized Asp824Asn (*Abe et al., 2018*) mutant implies that the formation of a salt bridge between Lys791 and Asp824 drives the transport cycle forward from E2P. Another acidic side chain Asp942 (TM8) makes a salt bridge with Arg946 (3.3 Å), and is therefore likely to be deprotonated, despite being rather distant from the $K^+$-binding site. Although the p$K_a$ of Asp942 is 7.6 in the crystal structure, it drops to 5.4 ± 0.02, 4.8 ± 0.02 and 5.4 ± 0.02 in the last 50 ns of three runs of the Tyr799Trp MD simulation, supporting the idea that this residue can be considered to be deprotonated. The numbers are qualitatively similar for the wild-type simulations. Arg946 is replaced with Cys937 near the third $Na^+$-binding site in $Na^+,K^+$-ATPase (*Kanai et al., 2013*), and therefore the Asp942–Arg946 salt bridge observed in the $H^+,K^+$-ATPase structure can be predicted to be related to the pump's electroneutral transport properties (*Holm et al., 2017*), although the function of the bridge is unclear in the absence of a high-resolution E1 structure. On the basis of these observations, we conclude that Glu936 is protonated, and Asp824 and Asp942 are deprotonated. We further evaluated the protonation states of the $K^+$-coordinating glutamate residues (Glu343, Glu795 and Glu820) by launching MD simulations of the pump with different protonation state combinations (*Figure 6*, *Table 3*).

The calculated average valence from each 50 ns (latter half of 100 ns simulation)×3 copies of MD trajectories indicates that the protonation state expected from the crystal structure, namely, protonated Glu343, Glu795 and Glu936 (*Figure 6*, for E343p/E795p/E936p) has a mean valence for $K^+$ closest to the ideal value (1.07) in all examined simulation set-ups (*Table 3*). During the simulation, $K^+$ was stably coordinated at the cation-binding site, and the root mean squared deviation (RMSD) of the ion remained below 1.0 Å in the probable protonation states. The calculated p$K_a$ values for Glu343, Glu795 and Glu820 in the static crystal structure are 8.5, 11.1 and 10.8, respectively,

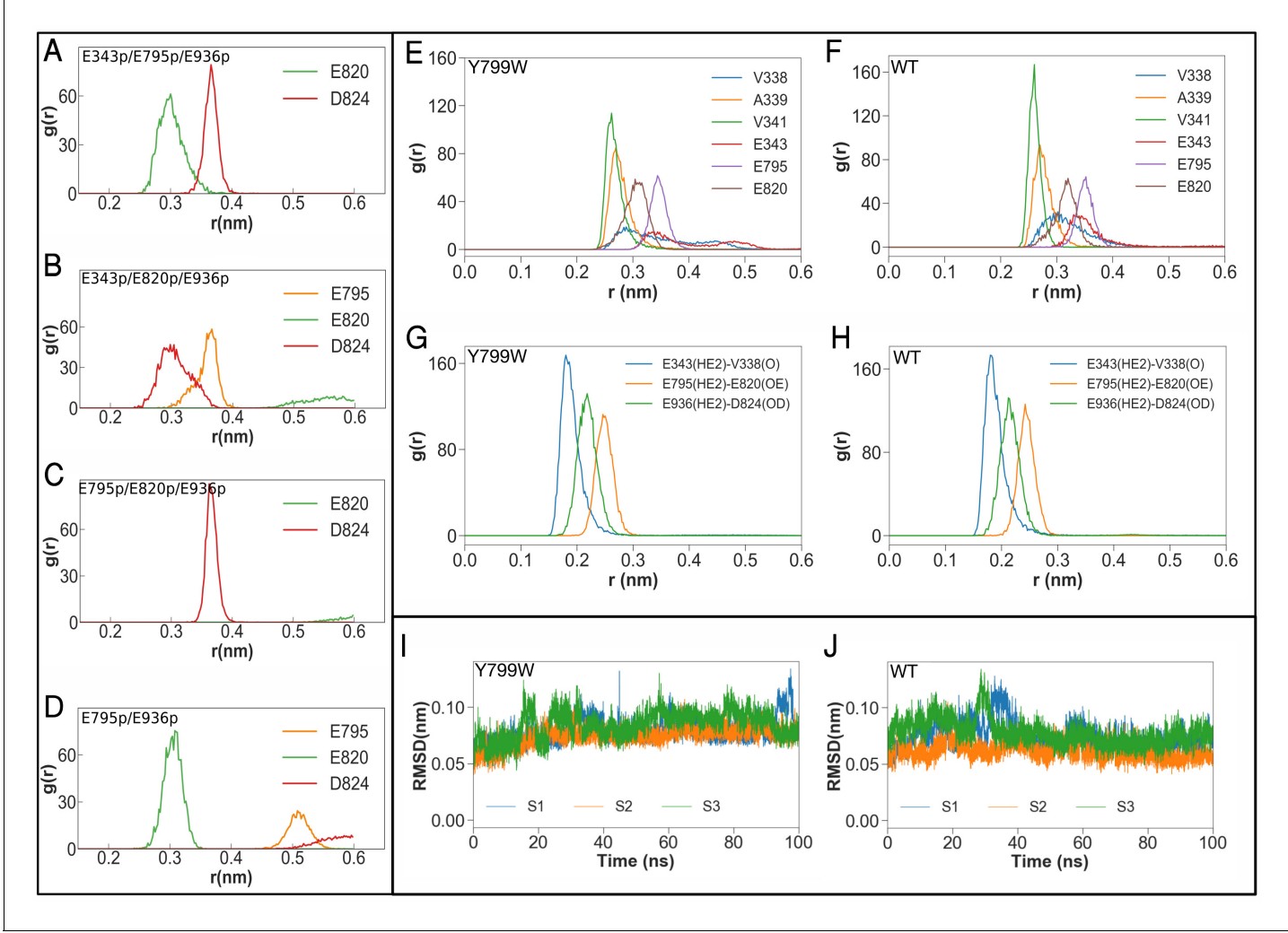

**Figure 6.** Molecular dynamics simulations of the $K^+$-binding site of wild-type $H^+,K^+$-ATPase (WT) and the Tyr799Trp (Y799W) mutant. (**A**, **B**, **C** and **D**) Salt-bridge formation with K791: the radial distribution function (RDF) is a normalized histogram of distances between the ε-amino group of Lys791 and the center of mass of the indicated side chain carboxylate oxygen atoms of acidic residues (r(nm)), as determined during the MD simulations with the indicated protonation combinations for the Y799W mutant. The protonated residues are marked 'p'. For example E795 and E936 are protonated in the simulation E795p/E936p, while the other three residues are deprotonated. Protonation status for D824, E936 and D942 remains the same (D824⁻/E936p/D942–) for these four simulations, and is not indicated in the figure. (**E**, **F**) RDFs between the $K^+$ ion and the oxygen atoms coordinating the ion for Y799W and WT. For acidic residues, the RDF is calculated between the $K^+$ ion and the center of mass of the two side chain carboxyl oxygen atoms. For the non-acidic residues, the carbonyl oxygen atom is used. (**G**, **H**) RDFs calculated between protonated side chains protons of Glu343 (E343p), Glu795 (E795p), Glu936 (E936p) with their hydrogen bonding acceptors on Val338 (V338), Asp820 (E820) and Asp824 (D824), respectively. All RDFs were calculated from a concatenated trajectory containing the last 50 ns of the three simulation copies of each system. (**I**, **J**) The RMSD of the amino acids in the binding site in three independent simulations with different initial velocity distributions.

DOI: https://doi.org/10.7554/eLife.47701.014

suggesting that the protonation status of E343p/E795p/E936p expected from the Tyr799Trp crystal structure is the most likely.

The salt bridge between Lys791 and Glu820 is a key feature of the cation-binding site of the $H^+$, $K^+$-ATPase ($K^+$)E2-P state. We quantified this salt-bridge in MD simulations by calculating RDFs between the ε-amino group of Lys791 with surrounding acidic side chains (*Figure 6A–D*). RDF plots of Lys791 in E343p/E795p/E936p and E795p/E936p show sharp distributions with Glu820 (indicating the formation of stable salt bridges between them during simulations), which contrast markedly with the distribution predicted when a protonated Glu820 is assumed. However, the calculated valence for E795p/E936p is significantly higher (1.27) than that of E343p/E795p/E936p (*Table 3*). We

**Table 3.** Mean valences calculated in the K⁺-binding site of H⁺,K⁺-ATPase, assuming protonation for the acidic residues as evaluated by MD simulations.

| | Tyr799Trp | | | | | | | | WT |
|---|---|---|---|---|---|---|---|---|---|
| | 2-proton states | | | 3-proton states | | | 4-proton states | | |
| E343 | $H^+$ | – | – | $H^+$ | $H^+$ | – | $H^+$ | $H^+$ | $H^+$ |
| E795 | – | $H^+$ | – | $H^+$ | – | $H^+$ | $H^+$ | – | $H^+$ |
| E820 | – | – | $H^+$ | – | $H^+$ | $H^+$ | – | $H^+$ | – |
| D824 | – | – | – | – | – | – | $H^+$ | $H^+$ | – |
| E936 | $H^+$ | $H^+$ | $H^+$ | $H^+$ | $H^+$ | $H^+$ | $H^+$ | – | $H^+$ |
| D942 | – | – | – | – | – | – | – | $H^+$ | – |
| | Valence† (Mean ± SEM) | | | | | | | | |
| V338 O | 0.077 ± 0.001 | 0.239 ± 0.004 | 0.282 ± 0.002 | 0.042 ± 0.002 | 0.116 ± 0.004 | 0.268 ± 0.001 | 0.111 ± 0.003 | 0.217 ± 0.001 | 0.074 ± 0.003 |
| A339 O | 0.132 ± 0.001 | 0.098 ± 0.002 | 0.005 ± 0.000 | 0.171 ± 0.004 | 0.235 ± 0.002 | 0.128 ± 0.001 | 0.203 ± 0.003 | 0.248 ± 0.001 | 0.190 ± 0.003 |
| V341 O | 0.242 ± 0.001 | 0.278 ± 0.005 | 0.224 ± 0.001 | 0.239 ± 0.003 | 0.269 ± 0.001 | 0.283 ± 0.002 | 0.295 ± 0.003 | 0.29 ± 0.001 | 0.283 ± 0.003 |
| E343 Oε1 | 0.21 ± 0.001 | 0.169 ± 0.004 | 0.257 ± 0.002 | 0.046 ± 0.002 | – | 0.26 ± 0.002 | 0.068 ± 0.003 | – | 0.091 ± 0.001 |
| E343 Oε2 | 0.005 ± 0.000 | 0.152 ± 0.003 | 0.284 ± 0.002 | 0.002 ± 0.000 | – | 0.206 ± 0.002 | 0.002 ± 0.000 | – | 0.003 ± 0.001 |
| E795 Oε1 | 0.278 ± 0.003 | 0.121 ± 0.002 | 0.291 ± 0.002 | 0.267 ± 0.003 | 0.315 ± 0.002 | 0.107 ± 0.001 | 0.277 ± 0.003 | 0.239 ± 0.003 | 0.236 ± 0.003 |
| E795 Oε2 | 0.134 ± 0.003 | –‡ | 0.272 ± 0.002 | – | 0.053 ± 0.002 | 0.001 ± 0.000 | – | 0.239 ± 0.003 | 0.002 ± 0.000 |
| E820 Oε1 | 0.200 ± 0.002 | 0.017 ± 0.001 | 0.001 ± 0.000 | 0.015 ± 0.000 | 0.061 ± 0.001 | 0.001 ± 0.000 | 0.138 ± 0.005 | 0.155 ± 0.001 | 0.018 ± 0.000 |
| E820 Oε2 | 0.109 ± 0.002 | 0.195 ± 0.004 | 0.010 ± 0.000 | 0.291 ± 0.006 | 0.089 ± 0.001 | 0.028 ± 0.000 | 0.118 ± 0.005 | 0.066 ± 0.001 | 0.273 ± 0.003 |
| total | 1.387 | 1.269 | 1.626 | 1.073¶ | 1.138 | 1.282 | 1.212 | 1.454 | 1.171 |

†Ideal value for K⁺ is 1.00. ‡Only oxygen atoms within 4Å of bound K⁺ were included for the valence calculation. ¶The most likely protonation state of E343p/E795p/E936p shows a total valence closest to the ideal value amongst all simulations evaluated.

DOI: https://doi.org/10.7554/eLife.47701.015

therefore conclude that E343 is probably protonated, as expected from the crystal structure. Accordingly, Glu820 is the only deprotonated acidic residue that coordinates bound K⁺. However, the negative charge of Glu820 is neutralized by a stable salt bridge with Lys791 over the entire simulation period (as seen in the RDF calculations), which is not observed at all in the simulations assuming Glu820 protonation (*Figure 3*, *Figure 6*). Therefore, these analyses assuming different combinations of protonated states are consistent with the protonation status expected from the crystal structure, namely, E343p/E795p/E936p.

To further compare the cation-binding site structure of the wild-type enzyme and the Tyr799Trp mutant, we launched MD simulations of both systems. RDFs between the K⁺ ion and the coordinating oxygen atoms (*Figure 6E–F*) show that the ion coordination geometry of the wild-type is very similar to that of the mutant. The only minor differences are with respect to the side chain of residue Glu343 and the backbone carbonyl of Val 338, both of which coordinate the ion better in the wild-type. Furthermore, the RMSD of the ion-binding residues (*Figure 6I–J*) from the initial crystal structure is nearly identical for the wild-type enzyme and the Tyr799Trp mutant, indicating a stable coordination geometry in both systems. We observe tight coordination between Glu343p and Val341, Glu795p and Glu820, and also Glu936p and Asp824 in MD simulations of the E343p/E795p/E936p system for both the wild-type and the Tyr799Trp mutant (*Figure 6G–H*). These analyses further support the conclusion that the results obtained from MD simulation of Tyr799Trp can be extended to the wild-type enzyme.

## Discussion

On the basis of the crystal structure and the protonation status supported by MD simulation, the previously proposed transport model for H⁺,K⁺ transport by H⁺,K⁺-ATPase (*Abe et al., 2018*) needs to be revised (*Figure 7*). In the K⁺-occluded E2-P transition state, a single K⁺ is observed at the cation-binding site of both the wild-type enzyme and the Tyr799Trp mutant (*Figure 3*). Assuming

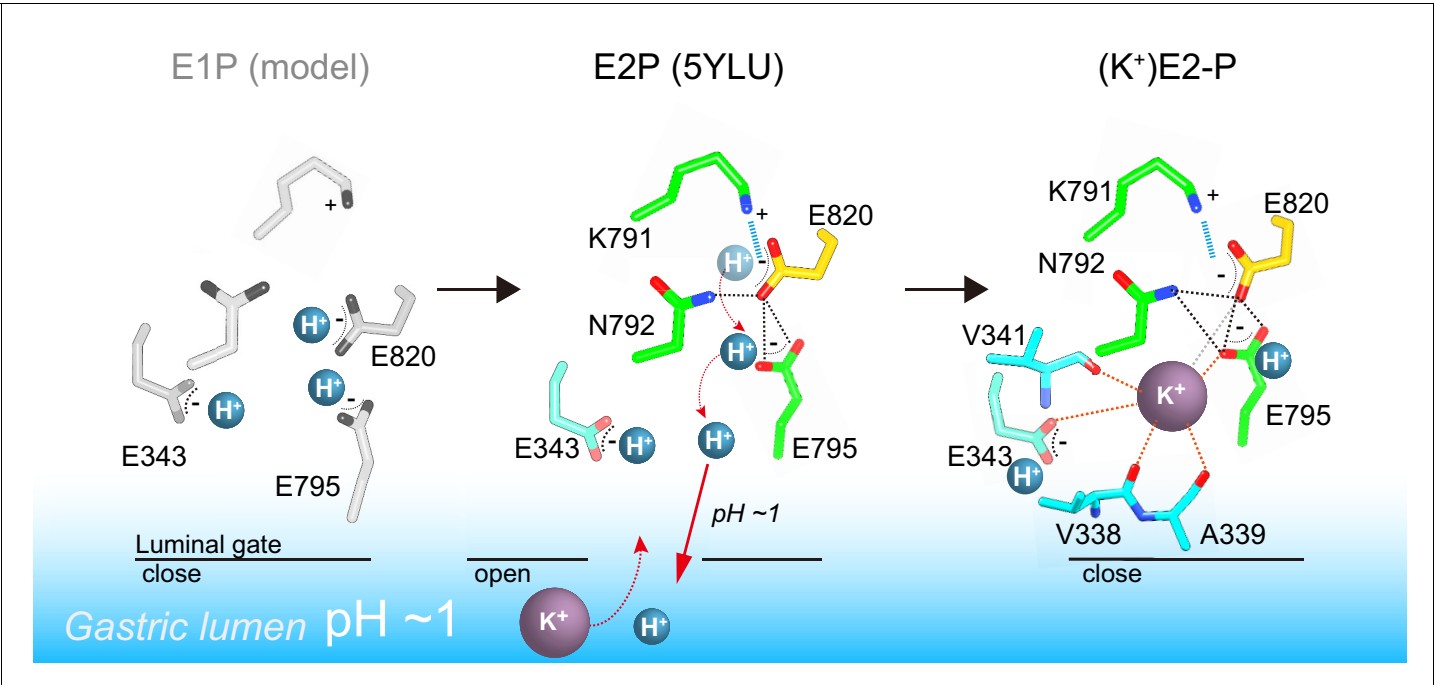

**Figure 7.** A model of one H⁺ and one K⁺ transport by H⁺,K⁺-ATPase. In the H⁺-occluded E1P state (left), all of the acidic residues are expected to be protonated. After a single H⁺ is extruded to the gastric luminal solution in the luminal-open E2P state (center), a single K⁺ is occluded in the following luminal-closed (K⁺)E2-P transition state (right). Dotted lines indicate hydrogen bonds (black), a salt bridge (blue), and K⁺-coordination by oxygen atoms (orange, ≤3 Å; gray >3 Å).

DOI: https://doi.org/10.7554/eLife.47701.016

electroneutral transport (*Sachs et al., 1976*; *van der Hijden et al., 1990*; *Burnay et al., 2001*; *Burnay et al., 2003*), one H⁺ must be exported in the prior luminal-open E2P state. Extensive hydrogen bonds and a salt-bridge network centered on Glu820 in the luminal-open E2P state ensure that only one H⁺ is likely to be extruded from Glu820, regardless of the luminal pH (*Abe et al., 2018*). Accordingly, even when the luminal solution is at neutral pH, one H⁺ will remain bound to Glu343, as suggested by the crystal structures (*Figure 5*) and supported by the MD simulations (*Figure 6*).

In the cation-binding site of the H⁺,K⁺-ATPase structure, there is not enough space to accommodate a second K⁺ at the position corresponding to site I of Na⁺,K⁺-ATPase (*Figure 5C,D*, *Figure 8*). This restricted structure is due to the position of the Lys791 side chain and that of its salt-bridge partner Glu820. These two amino acids are invariant for the gastric α1 isoform of H⁺,K⁺-ATPase, but are replaced with Ser782 and Asp811 (shark α1 sequence), respectively, in Na⁺,K⁺-ATPase (*Figure 8—figure supplement 1*). In particular, Ser782 in Na⁺,K⁺-ATPase plays an important role in K⁺-accommodation at site I (*Morth et al., 2007*; *Shinoda et al., 2009*). Replacing Ser with the bulky and positively charged Lys791 in H⁺,K⁺-ATPase would prevent K⁺ binding at a site I position. In fact, when the two structures (H⁺,K⁺-ATPase and Na⁺,K⁺-ATPase) were superimposed, K⁺ at site I and a coordinating water molecule in Na⁺,K⁺-ATPase sterically clashed with the H⁺,K⁺-ATPase Lys791 side chain (*Figure 7*), the position of which is defined by its salt-bridge partner Glu820 and by a hydrogen bond with the main chain carbonyl of Thr788 (*Figure 5*). In contrast to the important contribution of the side chain carbonyl of Asn783 for K⁺ coordination in Na⁺,K⁺-ATPase (*Shinoda et al., 2009*), the carbonyl oxygen of the Asn792 side chain in H⁺,K⁺-ATPase faces towards the opposite side of K⁺, and the Asn792 amino group stabilizes the positions of Glu795 and Glu820 by making hydrogen bonds (*Figure 5C,D*). In the Na⁺,K⁺-ATPase structure, Asn783 is stabilized by forming a hydrogen bond with Tyr854 in TM7 (*Shinoda et al., 2009*). In H⁺,K⁺-ATPase, because of Tyr340 (corresponding to Asn331 in Na⁺,K⁺-ATPase) in TM4, the side chain of Tyr863 (TM7, corresponds to Tyr854 in Na⁺,K⁺-ATPase) is removed from the position observed in Na⁺,K⁺-ATPase, which brings a

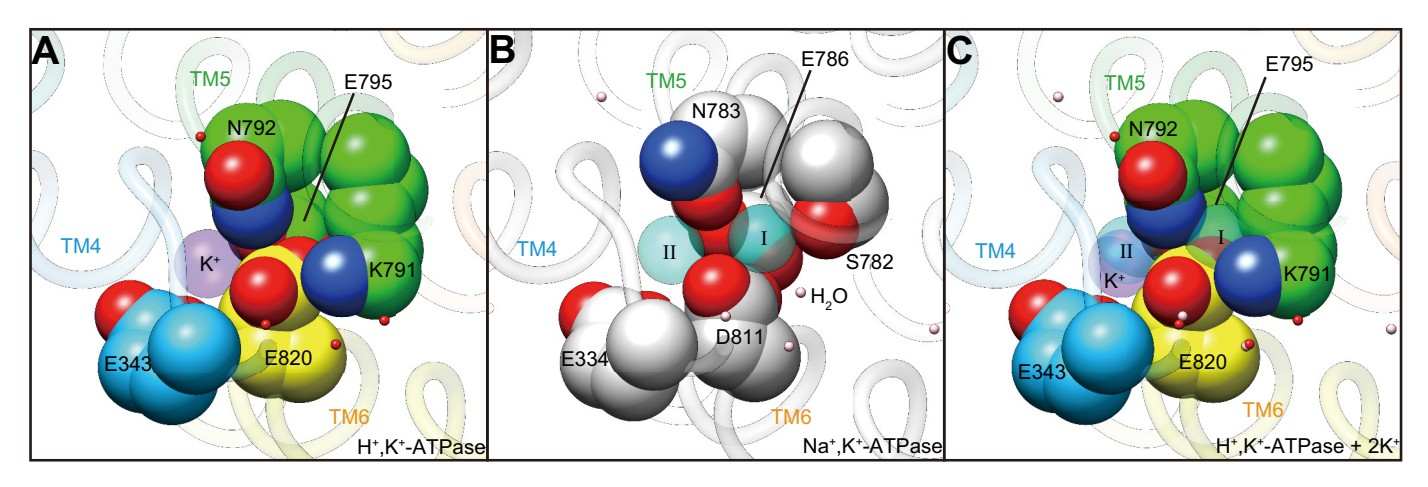

**Figure 8.** Comparison of the cation-binding site between $H^+,K^+$-ATPase and $Na^+,K^+$-ATPase. Cation-binding sites of $H^+,K^+$-ATPase Y799W($K^+$)E2-MgF$_x$ (A) and $Na^+,K^+$-ATPase ($2K^+$)E2-MgF$_x$ (B) are shown. Two $K^+$ ions that are occluded in $Na^+,K^+$-ATPase (I, II) were superimposed on the $H^+,K^+$-ATPase structure (C), showing significant steric clash between $K^+$ at site I, and Lys791 and Glu820. Only side chains that are important for $K^+$ coordination are displayed as space-fill models. Water molecules in $H^+,K^+$-ATPase and $Na^+,K^+$-ATPase are shown as red and pink dots, respectively. Figures are viewed from approximately perpendicular to the membrane from the cytoplasmic side. Color codes as in **Figure 3**.
DOI: https://doi.org/10.7554/eLife.47701.017

The following figure supplement is available for figure 8:

**Figure supplement 1.** Sequence alignment of TM helices of $H^+,K^+$–ATPases and $Na^+,K^+$-ATPases.
DOI: https://doi.org/10.7554/eLife.47701.018

water molecule in between Asn792 and Tyr863 in $H^+,K^+$-ATPase, and stabilizes the Asn792 side chain. Except for the observed differences described above, the cation-binding site of these two related ATPases is surprisingly similar, with respect not only to the positions of the side chains but also to those of water molecules.

It is noteworthy that single $K^+$ binding is driven by the Lys791–Glu820 interaction (**Figures 5** and **8**), which also plays a key role in $H^+$ extrusion into the acidic gastric solution (**Abe et al., 2018**). The salt bridge is required for $H^+$ extrusion, but it also prevents binding of a second $K^+$. Thus, $H^+,K^+$-ATPase seems to sacrifice $2H^+/2K^+$ transport in order to achieve the energetically challenging $H^+$ transport of over a million-fold $H^+$ gradient. The single $K^+$ binding structure of $H^+,K^+$-ATPase therefore represents a remarkable example of how energetic barriers faced by membrane pumps are overcome in living systems. Evolutionary pressure selected single $H^+/K^+$ transport, rather than the more efficient double transport mode of $2H^+/2K^+$, in order to achieve the thermodynamically challenging task of $H^+$ uptake against a pH 1 solution in the stomach.

## Materials and methods

### Protein expression and purification

Procedures for protein expression and purification are essentially the same as those reported previously (**Abe et al., 2018**). Briefly, the wild-type $H^+,K^+$-ATPase (WT) or a Tyr799Trp (Y799W) mutant of the $H^+,K^+$-ATPase αβ-complex was expressed in the plasma membrane using baculovirus-mediated transduction of mammalian HEK293S GnT1$^-$ cells purchased from ATCC (**Goehring et al., 2014**). Cells were not tested for mycoplasma contamination. The harvested cells were broken up using a high-pressure emulsifier (Avestin), and membrane fractions were collected. Membrane fractions were solubilized with 1% octaethylene glycol monododecyl ether ($C_{12}E_8$, Nikko Chemical) with 40 mM MES/Tris (pH 6.5), 10% glycerol, 5 mM dithiothreitol in the presence of 50 mM $CH_3$COORb, 10 mM $MgCl_2$, 10 mM NaF for WT(Rb$^+$)E2-MgF, or 200 mM KCl, 10 mM $MgCl_2$, 10 mM NaF for Y799W($K^+$)E2-MgF, or 200 mM RbCl, 10 mM $MgCl_2$, 10 mM NaF for Y799W(Rb$^+$)E2-MgF or 200 mM RbCl,

1 mM $MgCl_2$, 1 mM $AlCl_3$, 4 mM NaF for $Y799W(Rb^+)E2-AlF$, on ice for 20 min. Proteins were affinity purified by anti-Flag M2 affinity resin (Sigma-Aldrich), which followed digestion of the affinity tag and deglycosilation by TEV protease and MBP-fusion endoglycosidase (New England Biolabs) at 4°C overnight. Samples were further purified by a size-exclusion column chromatograph using a Superose6 Increase column (GE Healthcare). Peak fractions were collected and concentrated to 10 mg/ml. The concentrated $H^+,K^+$-ATPase samples were added to the glass tubes in which a layer of dried dioleoyl phosphatidylcholine had formed, in a lipid-to-protein ratio of 0.1–0.4, and incubated overnight at 4°C in a shaker mixer operated at 120 rpm. After removing the insoluble materials by ultracentrifugation, the lipidated samples were used for the crystallization.

## Crystallization

Initial screening was performed using a $K^+$ salt-containing matrix called $K^+$ night screen, which was developed on the basis of the King screen (*Gourdon et al., 2011*). Crystals were obtained by vapor diffusion at 20°C. For the Y799W mutant, a 5 mg/ml purified, lipidated protein sample was mixed with a reservoir solution containing 10% glycerol, 20% PEG2000MME, 3% methylpentanediol, and 5 mM β-mercaptoethanol in the presence of 0.4 M KCl for the $Y799W(K^+)E2-MgF_x$state, or 0.4 M RbCl for the $Y799W(Rb^+)E2-MgF_x$ and $Y799W(Rb^+)E2-AlF_x$ states. For the WT enzyme, reservoir solution containing 10% glycerol, 15% PEG6000, 0.1 M $CH_3COORb$, 6% methylpentanediol and 5 mM β-mercaptoethanol was used. Crystals were flash frozen in liquid nitrogen.

## Structural determination and analysis

Diffraction data were collected at the SPring-8 beamline BL32XU and BL41XU, and processed using XDS. Structure factors were subjected to anisotropy correction using the UCLA MBI Diffraction Anisotropy server (*Strong et al., 2006*) (http://services.mbi.ucla.edu/anisoscale/). The structure of $Y799W(Rb^+)E2-AlF_x$ was determined by molecular replacement with PHASER, using an atomic model of $H^+,K^+$-ATPase in the SCH28080-bound E2BeF state (pdb ID: 5YLV) as a search model. Coot (*Emsley and Cowtan, 2004*) was used for cycles of iterative model building and Refmac5 and Phenix (*Adams et al., 2010*) were used for refinement. Other structures described in this paper were determined by molecular replacement using an atomic model of $Y799W(Rb^+)E2-AlF_x$ state as a search model. Rubidium and potassium ions were identified in anomalous difference Fourier maps calculated using data collected at wavelengths of 0.8147 Å and 1.700 Å, respectively. The $Y799W(K^+)E2-MgF_x$, $Y799W(Rb^+)E2-MgF_x$, $Y799W(Rb^+)E2-AlF_x$ and $WT(Rb^+)E2-MgF_x$ models contained 98.2/1.8/0.0%, 98.3/1.7/0.0%, 98.2/1.8/0.0% and 91.1/8.8/0.1% in the favored, allowed, and outlier regions of the Ramachandran plot, respectively.

## Activity assay using recombinant proteins

The wild-type or mutant α-subunit was co-expressed with the wild-type β-subunit using the BacMam system as described above, and broken membrane fractions were collected. $H^+,K^+$-ATPase activity was measured as described previously (*Abe et al., 2017*). Briefly, permeabilized membrane fractions (wild-type or mutant) were suspended in buffer comprising 40 mM PIPES/Tris (pH 7.0), 2 mM $MgCl_2$, 2 mM ATP, and 0–50 mM KCl in the presence of three different concentrations of vonoprazan, or their absence, in 96-well plates. Reactions were initiated by incubating the fractions at 37°C using a thermal cycler, and maintained for 1 to 5 hr depending on their activity. Reactions were terminated, and the amount of released inorganic phosphate was determined colorimetrically using a microplate reader (TECAN).

## Molecular dynamics simulations

All simulations were performed using GROMACS (v-2016.3) (*Abraham et al., 2015*; *Berendsen et al., 1995*; *Hess et al., 2008*; *Pronk et al., 2013*; *Van Der Spoel et al., 2005*). The CHARMM36 force field (v-July 2017) (*Bjelkmar et al., 2010*; *Klauda et al., 2010*; *MacKerell et al., 1998*; *Mackerell et al., 2004*) was used to model all the components of the system. A symmetric lipid bilayer containing ~500 DOPC lipids was generated using CHARMM-GUI (*Jo et al., 2009*; *Lee et al., 2016*). An atomic model of $H^+, K^+$-ATPase $Y799W(K^+)E2-MgF_x$ was placed in the bilayer using the OPM (*Lomize et al., 2006*) server. All the components of the experimentally derived structure were kept intact in the system except detergent and $MgF_4^{2-}$ which was replaced with a $PO_4^{3-}$

molecule. Systems were hydrated with ~81,000 TIP3P water molecules and ionized with 150 mM NaCl.

Neighbor search for non-bonded interactions was performed within a cutoff of 12 Å and revised after every 20 steps. Van der Waals forces were smoothly switched off between 10 Å and 12 Å with a force-switch function. The particle-mesh Ewald (PME) (*Darden et al., 1993*; *Essmann et al., 1995*) method was used to calculate electrostatic interactions. Initially, an energy minimization was performed using the gradient descent method to remove steric clashes. Then, systems were equilibrated for a total of 25–30 ns in three steps. In the first step, heavy atoms of the protein including the potassium ions were restrained with a force constant of 1000 kJ/mol·nm$^2$, which was reduced to zero in two further equilibration steps. Finally, a production run was performed for 100 ns without restraints. Periodic boundary conditions were incorporated. All acidic residues pointing towards the stomach lumen were kept protonated. Overall, 18 protonation states with either two, three or four acidic protonated residues were simulated. For some protonation states that were deemed more likely, on the basis of the analysis, we launched two further copies of the simulations with different initial velocity distributions. The results from all simulation copies are nearly identical. A time-step of 1 fs was used for the simulations. The temperature of the systems was stabilized at 310 K by integrating the Nosé-Hoover thermostat (*Hoover, 1985*; *Nosé, 1984*) during the production runs. The Parrinello-Rahman barostat (*Parrinello and Rahman, 1981*), along with a semi-isotropic pressure coupling scheme, was used to maintain the pressure at 1 bar. All bonds containing hydrogen were constrained using The Linear Constraint Solver algorithm (LINCS) (*Hess et al., 1997*). PROPKA (v-3.1) (*Li et al., 2005*; *Olsson et al., 2011*; *Søndergaard et al., 2011*) was used to measure p$K_a$ in the crystal structure. Only the last 50 ns were considered for the data analysis. GROMACS and python scripts were used for the various analyses used in the manuscript. Snapshots were generated using Visual Molecular Dynamics (VMD) (*Humphrey et al., 1996*).

## Acknowledgements

We thank M Taniguchi for technical assistance; D McIntosh for improving the manuscript; and C Toyoshima for discussions. The synchrotron radiation experiments were performed at BL32XU and BL41XU in SPring-8 with the approval of the Japan Synchrotron Radiation Research Institute (JASRI Proposal numbers: 2017B2701 and 2018B2703). We thank the beamline staff for their facilities and support. The simulations were carried out at the Danish e-Infrastructure Cooperation (DeiC) National HPC Center, on ABACUS 2.0 at the University of Southern Denmark, SDU, and on computing resources on the Swiss cluster Piz Daint as part of a PRACE project (grant number 2016153468).

## Additional information

### Competing interests

Yoshinori Fujiyoshi: Is a director of CeSPIA Inc. The other authors declare that no competing interests exist.

### Funding

| Funder | Grant reference number | Author |
| --- | --- | --- |
| Japan Society for the Promotion of Science | 17H03653 | Kazuhiro Abe |
| Japan Science and Technology Agency | JPMJCR14M4 | Kazuhiro Abe |
| Japan Agency for Medical Research and Development | BINDS | Kazuhiro Abe |
| Takeda Science Foundation | | Kazuhiro Abe |
| New Energy and Industrial Technology Development Organization | | Yoshinori Fujiyoshi |

| Japan Agency for Medical Research and Development | Yoshinori Fujiyoshi |
|---|---|
| Lundbeckfonden | Vikas Dubey Himanshu Khandelia |

The funders had no role in study design, data collection and interpretation, or the decision to submit the work for publication.

### Author contributions
Kenta Yamamoto, Formal analysis, Investigation, Writing—review and editing; Vikas Dubey, Data curation, Software, Formal analysis, Investigation, Writing—review and editing; Katsumasa Irie, Data curation, Formal analysis, Validation, Investigation, Writing—review and editing; Hanayo Nakanishi, Investigation, Writing—review and editing; Himanshu Khandelia, Software, Formal analysis, Supervision, Validation, Investigation, Writing—original draft, Writing—review and editing; Yoshinori Fujiyoshi, Funding acquisition, Project administration, Writing—review and editing; Kazuhiro Abe, Conceptualization, Resources, Data curation, Formal analysis, Supervision, Funding acquisition, Validation, Investigation, Visualization, Methodology, Writing—original draft, Project administration, Writing—review and editing

### Author ORCIDs
Katsumasa Irie https://orcid.org/0000-0002-8178-1552
Kazuhiro Abe https://orcid.org/0000-0003-2681-5921

### Decision letter and Author response
Decision letter https://doi.org/10.7554/eLife.47701.029
Author response https://doi.org/10.7554/eLife.47701.030

# Additional files

### Supplementary files
• Transparent reporting form
DOI: https://doi.org/10.7554/eLife.47701.019

### Data availability
Diffraction data have been deposited in PDB under accession code 6JXH, 6JXI, 6JXJ and 6JXK.

The following datasets were generated:

| Author(s) | Year | Dataset title | Dataset URL | Database and Identifier |
|---|---|---|---|---|
| Abe K, Irie K | 2019 | K+-bound E2-MgF state of the gastric proton pump (Tyr799Trp) | https://www.rcsb.org/structure/6JXH | Protein Data Bank, 6JXH |
| Abe K, Irie K | 2019 | Rb+-bound E2-MgF state of the gastric proton pump (Tyr799Trp) | https://www.rcsb.org/structure/6JXI | Protein Data Bank, 6JXI |
| Abe K, Irie K | 2019 | Rb+-bound E2-AlF state of the gastric proton pump (Tyr799Trp) | https://www.rcsb.org/structure/6JXJ | Protein Data Bank, 6JXJ |
| Abe K, Irie K | 2019 | Rb+-bound E2-MgF state of the gastric proton pump (Wild-type) | https://www.rcsb.org/structure/6JXK | Protein Data Bank, 6JXK |

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
