## [Decision Letter]

Thank you for submitting your article "A single K^+^-binding site in the crystal structure of the gastric proton pump" for consideration by *eLife*. Your article has been reviewed by three peer reviewers, including José D. Faraldo-Gómez as the Reviewing Editor, and the evaluation has been overseen by Olga Boudker as the Senior Editor. The following individuals involved in review of your submission have agreed to reveal their identity: Poul Nissen (Reviewer #2); Pablo Artigas (Reviewer #3).

The reviewers have discussed the reviews with one another and the Reviewing Editor has drafted this decision to help you prepare a revised submission.

Summary:

Abe and co-workers report structural and functional studies of the gastric H/K-ATPase, aimed at establishing the ion stoichiometry of the transport reaction. The central conclusion of these studies is that this pump binds and translocates only one K^+^ ion in each transport cycle. Because the activity of the pump is electroneutral, the implication of this result is that only one H^+^ is pumped into the gastric lumen in exchange for the K^+^ ion moving into the cytoplasm. This result is important for two reasons. It had been anticipated that the H/K-ATPase exchanges H^+^ and K^+^ in a 1:1 stoichometry in conditions whereby the pH in the gastric lumen is low; however, it had been previously proposed that the pump switches to a 2H^+^:2K^+^ exchange mode when that pH raises above a certain threshold. The notion that ion-coupled transport systems might vary their ion stoichiometry depending on conditions has been the subject of debate in the membrane transport field, to this date. Here, the authors contradict earlier work and show that the gastric H/K-ATPase has a fixed stoichiometry irrespective of the pH condition – namely 1H^+^:1K^+^. Secondly, the work underscores how subtle structural differences among seemingly related transporters have important functional consequences, and how these differences cannot be easily foreseen. The close homology between the H/K-ATPase and the Na/K-ATPase, which binds and translocates two K^+^ ions had been interpreted as supporting the view that the former can translocate 2 K^+^. However the structures now presented reveal that a close salt-bridge interaction unique to the H/K-ATPase blocks one of the cation binding at site in the Na/K-ATPase, explaining why this pump translocates 1 K^+^ ion.

The reviewers described these findings as "important" and "exciting" from a mechanistic standpoint. The structural biology was described as "elegant" and the functional data as "solid evidence". The computational component was, however, found to be unsuitable for *eLife* in its present form.

Essential revisions:

1) The manuscript needs to be extensively revised to improve its readability, and to highlight the significance of the results and provide more context. It is strongly recommended that the Results section is organized into subsections. Each section should have a clear focus e.g. stoichiometry, K^+^ coordination, conformational changes, etc., and start with a sentence or two that outline the question at hand, and end with another sentence or two that summarize what was learnt. The *eLife* format affords authors the opportunity to write their manuscripts with the readers in mind. The authors are urged to take advantage of this format. One a related note, the figures reporting functional data ought to be included in the main manuscript – particularly Supplementary figure 1.

2) The WT structure appears not to be well-determined and shows e.g. high R-factors (R_free_ 33.9%), which are not expected when a presumably near-identical high-resolution, template structure is available. It is a concern, therefore, that WT and mutant structures might differ in significant ways. The authors are asked to provide a more detailed description of the WT structure, despite its lower resolution, so as to clarify that the conclusions extracted for the Tyr799Trp mutant in regard to K^+^ stoichiometry can be extended to the WT protein as well. On a related note, the anomalous difference Fourier map for the WT data should be shown more clearly and at lower contour levels, ideally overlapping with sites I and II. It is important that the authors confirm that there is no indication of partial Rb^+^/K^+^ binding at site I, possibly coupled to a dynamic Glu-Lys salt bridge. (Note that 4-5 Å resolution is also where two closely spaced sites may merge into one peak, and in particular if phases are not very good).

3) The manuscript includes a computational component – which consists of molecular dynamics simulations of the new structure in a phospholipid membrane. In stark contrast to the experimental component, the quality and significance of this work does not measure up to the level expected for a journal such as *eLife*. We refer in particular to the simulations carried out to evaluate different protonation states, summarized in Figure 5 and Supplementary figure 6. The authors' strategy is to consider different combinations of protonation states for a specific set of acidic side-chains, including those in the vicinity of the K^+^ binding site, and carry out a simulation for each combination. Each simulation is about 100 ns long. The authors then extract snapshots from these simulations, and feed them to PROPKA, to "measure pKa dynamically". This approach, however, lacks any theoretical foundation. An evaluation of how the pKa of these side-chains arise from their dynamic interplay and interactions with solvent, K^+^, etc. would be indeed of interest; however, such analysis would require a different simulation design, e.g. a free-energy methodology with which to factor all contributions rigorously (see e.g. Huang et al. Nat. Commun. 2016; Marinelli et al. PNAS 2014). PROPKA is a heuristic tool to guide the interpretation of static structures, based on qualitative factors such as accessibility and proximity to other ionizable side-chains. To apply this tool to simulation snapshots does not imply that the calculated pKa values arise from the dynamics of the system, nor does it make PROPKA more reliable or sophisticated than when applied to a crystal structure. There is no such thing as a "pKa distribution"; whether these distributions are found to be unimodal or bimodal has dubious physical meaning. Ultimately, it is very much unclear whether this simulation analysis adds valuable insights beyond what can be inferred from the structure. Therefore, this element of the manuscript (i.e. Figure 5 and Supplementary figure 6, Results, tenth paragraph) must be removed altogether prior to resubmission. Instead, the authors should report PROPKA predictions for the structure itself, and comment on whether these predictions agree with their intuitive interpretation of the structure and existing mutational data (Results, seventh paragraph): i.e. that protons are bound to Glu343 (mediating a hydrogen-bond to the carbonyl of V338) and to Glu795 (mediating a hydrogen-bond to Glu820), as well as in between Glu936 to Asp824.

4) The simulations of the state with protonated Glu343, Glu795 and Glu936, for both WT and Y799W, are potentially informative but analyzed only superficially. Please provide a more detailed analysis beyond what is currently noted in regard to the coordination of the K^+^ ion (Results, ninth paragraph). For example, the authors should examine the stability and dynamics of the network of protein-protein, and protein-ion/water contacts outlined in Figure 3A, for both the WT and Y799W mutant, to highlight the most determinant of the geometry of this state; whether the transporter sustains an occluded state would also be of interest. Please discuss any potential mechanistic implications.

---

## [Author Response]

Essential revisions:1) The manuscript needs to be extensively revised to improve its readability, and to highlight the significance of the results and provide more context. It is strongly recommended that the Results section is organized into subsections. Each section should have a clear focus e.g. stoichiometry, K^+^ coordination, conformational changes, etc., and start with a sentence or two that outline the question at hand, and end with another sentence or two that summarize what was learnt. The eLife format affords authors the opportunity to write their manuscripts with the readers in mind. The authors are urged to take advantage of this format. One a related note, the figures reporting functional data ought to be included in the main manuscript – particularly Supplementary figure 1.

Thank you very much for giving us a chance to extensively re-write our manuscript. We now follow the *eLife* format and include more details in each section. The Results section is organized into subsections with short titles provided, and we have rearranged the text and figures. I especially appreciate the comment about functional studies. I strongly believe that these analyses are informative when considering the enzyme’s function, and in some cases, more informative than the crystal structure itself. Accordingly, we moved previous Supplementary figures 1 and 3 to Figure 2 and 4, respectively.

2) The WT structure appears not to be well-determined and shows e.g. high R-factors (R_free_ 33.9%), which are not expected when a presumably near-identical high-resolution, template structure is available. It is a concern, therefore, that WT and mutant structures might differ in significant ways. The authors are asked to provide a more detailed description of the WT structure, despite its lower resolution, so as to clarify that the conclusions extracted for the Tyr799Trp mutant in regard to K^+^ stoichiometry can be extended to the WT protein as well. On a related note, the anomalous difference Fourier map for the WT data should be shown more clearly and at lower contour levels, ideally overlapping with sites I and II. It is important that the authors confirm that there is no indication of partial Rb^+^/K^+^ binding at site I, possibly coupled to a dynamic Glu-Lys salt bridge. (Note that 4-5 Å resolution is also where two closely spaced sites may merge into one peak, and in particular if phases are not very good).

I appreciate this very productive and educational comment about the WT structure. As the reviewers have realized, what we ultimately want to know is how the WT, and not the mutant enzyme, works. We use the mutant for high-resolution structural analysis, because of the limitations of WT crystals. So it is very important to describe the similarities and differences between the WT and mutant structures, although they are almost identical. According to the reviewers’ suggestion, now we provide an additional figure (Figure 3—figure supplement 2) to describe the WT structure extensively, and compare it with the high-resolution mutant structure. We again confirmed that the WT structure is really close to that of the Y799W mutant, and essentially the same with respect to the luminal gate conformation, azimuthal position of cytoplasmic domains, and number of K^+^/Rb^+^ (one) occluded. We previously judged our R_free_ value as acceptable, because it is comparable to other P-type ATPase structures with similar resolution range (e.g., 6HXB, SERCA2a, Rf 0.354, Rw 0.310, analyzed at 4Å; 4XE5 bovine NaK-ATPase, Rf 0.336, Rw 0.322, at 3.9 Å; 2XBE SERCA E2BeF, Rf 0.327, Rw 0.293, at 3.8 Å). However, as the reviewer has pinpointed, it is true that there are low-resolution structures that are presumably modeled with a high-resolution template structure (e.g., 4RET, digoxin-NaK-ATPase, Rf 0.253, Rw 0.221 at 4 Å). The reason why our Rf value is relatively high is unclear. Possible reasons are as follows; 1. due to crystal packing, N-domain of protomer B shows weak density, due to sparse crystal contact; 2. Part of the N-terminal tail of the β-subunit is visible in the WT, in contrast to the mutant structure. Both points are now presented in the figure. However, even if these parts are different between WT and Y799W mutant, the molecular conformation, defined by the luminal gate and azimuthal positions of A- and P-domains and connecting loop structure, are essentially the same. We also present Rb^+^ anomalous density much clearer – confirming little density is observed at the position corresponding to site I of Na/K-ATPase even at lower contour level. This makes our results much more reliable. I therefore again appreciate the reviewers’ comments on this issue.

3) The manuscript includes a computational component – which consists of molecular dynamics simulations of the new structure in a phospholipid membrane. […] Instead, the authors should report PROPKA predictions for the structure itself, and comment on whether these predictions agree with their intuitive interpretation of the structure and existing mutational data (Results, seventh paragraph): i.e. that protons are bound to Glu343 (mediating a hydrogen-bond to the carbonyl of V338) and to Glu795 (mediating a hydrogen-bond to Glu820), as well as in between Glu936 to Asp824.

We agree with the reviewer’s comment that pKa calculations from PROPKA are not as theoretically rigorous as, for example, a constant pH simulation. Instead, we now utilize the simulations under different protonation states to investigate the stability of the salt bridge and calculation of the total valence. We are confident that the simulations provide useful information about the stability of the salt bridge formed between E820 and K791 under different protonation conditions, because the same salt bridge is likely to be broken when the protonation state likely changes in the next conformation of the pumping cycle. Please also see the comment to point 4 below. We therefore keep the RDF plot and valence tables (as Table 3) provided by the MD simulation with different protonation states of acidic residues. We hope the reviewers agree with this idea.

4) The simulations of the state with protonated Glu343, Glu795 and Glu936, for both WT and Y799W, are potentially informative but analyzed only superficially. Please provide a more detailed analysis beyond what is currently noted in regard to the coordination of the K^+^ ion (Results, ninth paragraph). For example, the authors should examine the stability and dynamics of the network of protein-protein, and protein-ion/water contacts outlined in Figure 3A, for both the WT and Y799W mutant, to highlight the most determinant of the geometry of this state; whether the transporter sustains an occluded state would also be of interest. Please discuss any potential mechanistic implications.

According to the reviewers’ suggestion, we analyzed the WT and Y799W in more detail. The results are shown in different panels in Figure 6. The network of protein-protein and protein-ion interactions is compared carefully for the two systems, and minor differences are discussed.